# LAPLACIAN KERNELIZED BANDIT

**Shuang Wu**
Department of Statistics
UCLA
Los Angeles, 90095, USA
shuangwu2222@ucla.edu

**Arash A. Amini**
Department of Statistics
UCLA
Los Angeles, 90095, USA
aaamini@stat.ucla.edu

## ABSTRACT

We study multi-user contextual bandits where users are related by a graph and their reward functions exhibit both non-linear behavior and graph homophily. We introduce a principled joint penalty for the collection of user reward functions $\{f_u\}$, combining a graph smoothness term based on RKHS distances with an individual roughness penalty. Our central contribution is proving that this penalty is equivalent to the squared norm within a single, unified *multi-user RKHS*. We explicitly derive its reproducing kernel, which elegantly fuses the graph Laplacian with the base arm kernel. This unification allows us to reframe the problem as learning a single "lifted" function, enabling the design of principled algorithms, `LK-GP-UCB` and `LK-GP-TS`, that leverage Gaussian Process posteriors over this new kernel for exploration. We provide high-probability regret bounds that scale with an *effective dimension* of the multi-user kernel, replacing dependencies on user count or ambient dimension. Empirically, our methods outperform strong linear and non-graph-aware baselines in non-linear settings and remain competitive even when the true rewards are linear. Our work delivers a unified, theoretically grounded, and practical framework that bridges Laplacian regularization with kernelized bandits for structured exploration.

## 1 INTRODUCTION

Graphs are pervasive in modern sequential decision-making, encoding similarity or interaction among entities like users, items, or sensors. In a multi-user contextual bandit setting, this graph structure is informative since it provides a pathway to share information, allowing an algorithm to learn more efficiently than if it treated each user in isolation. We study the problem where a known user graph promotes *homophily*, meaning connected users tend to have similar reward functions. At each round $t$, a learner observes a user $u_t$ and a set of available arms (contexts) $\mathcal{D}_t \subset \mathbb{R}^d$, selects an arm $x_t \in \mathcal{D}_t$, and receives a noisy reward $y_t$. Naively learning a separate model for each user is inefficient, leading to regret that scales with the number of users. Exploiting the graph structure, however, can yield dramatic improvements in both sample efficiency and performance Szorenyi et al. (2013); Landgren et al. (2016); Gong & Zhang (2025); Wang et al. (2025).

This problem was first formalized as the Gang of Bandits (GOB) Cesa-Bianchi et al. (2013), which models the collection of user reward functions $\{f_u(\cdot)\}_{u=1}^n$ as a *smooth signal* on the graph. Seminal works like `GoB.Lin` Cesa-Bianchi et al. (2013) assume linear reward functions, $f_u(x) = \theta_u^\top x$, and penalize roughness via the graph Laplacian, leading to the effective linear bandit solution. Subsequent research has extended this approach with improved computational scaling Vaswani et al. (2017); Yang et al. (2020), but has largely remained within the linear paradigm. Yet, in many applications, from recommendation systems to personalized medicine, reward functions exhibit complex, non-linear behavior. While a rich literature on kernelized bandits exists to handle non-linear rewards for a single agent Chowdhury & Gopalan (2017); Du et al. (2021); Bubeck et al. (2021); Li et al. (2022); Zhou & Ji (2022), principled methods for the multi-user graph setting are less developed. Existing approaches construct a multi-user kernel heuristically as a product of user and arm kernels Dubey et al. (2020), leaving a gap between the intuitive modeling goal and the final algorithm. We refer to Appendix A for further discussion of the related work.

Our work bridges this gap, starting from a natural first principle for this problem. A desirable collection of reward functions $\{f_u\}_{u=1}^n$, where each $f_u$ lies in a Reproducing Kernel Hilbert Space (RKHS) $\mathcal{H}_x$, should be jointly regularized: they should be smooth across the graph (homophily) and individually well-behaved (low complexity). We formalize this via an intuitive, additive penalty that combines a graph smoothness term with a standard ridge penalty. In the scalar case without arm features, this type of Laplacian-based regularization is known to induce a kernel whose matrix is the (regularized) Green's function of the graph Smola & Kondor (2003). Building on this connection, we show that the same principle extends to the multi-user contextual setting: the joint penalty defines the squared norm of a single, lifted function $f(\boldsymbol{x}, u) := f_u(\boldsymbol{x})$ in a unified multi-user RKHS. We explicitly derive the reproducing kernel for this space, which elegantly fuses the graph Laplacian $L$ and the base arm kernel $K_x$: $K((\boldsymbol{x}, u), (\boldsymbol{x}', u')) = [\boldsymbol{L}_\rho^{-1}]_{u,u'} K_x(\boldsymbol{x}, \boldsymbol{x}')$, where $\boldsymbol{L}_\rho = \boldsymbol{L} + \rho \boldsymbol{I}$ is the regularized Laplacian.

This unifying perspective transforms the problem of learning $n$ related functions into the elegant problem of learning a single function in a well-defined kernel space. It allows us to directly apply the powerful machinery of Gaussian Process (GP) bandits Srinivas et al. (2009); Krause & Ong (2011); Vakili et al. (2021). We develop `LK-GP-UCB` and `LK-GP-TS`, algorithms whose principled uncertainty estimates are derived from the GP posterior of this unified kernel, enabling them to naturally and jointly leverage non-linear arm structure and Laplacian homophily. We provide regret guarantees for these algorithms in terms of an *effective dimension* that captures the spectral interplay between the graph and the kernel. Our experiments show that our methods are competitive in linear regimes and substantially outperform both linear and non-graph-aware baselines when rewards are non-linear yet graph-smooth.

Our main contributions are:

- We formalize the generalized gang-of-bandits problem with a principled joint penalty combining graph smoothness and RKHS regularity for the collection of reward functions $\{f_u\}_{u=1}^n$.
- We prove that this penalty is equivalent to the squared norm in a single multi-user RKHS and explicitly derive its reproducing kernel, unifying the graph and arm structures.
- We develop `LK-GP-UCB` and `LK-GP-TS`, GP-based bandit algorithms that leverage this unified kernel for principled and effective exploration.
- We provide novel regret bounds in terms of an effective dimension that depends on the spectral properties of both the kernel and the graph Laplacian.
- We empirically validate our approach, demonstrating significant performance gains over strong baselines in settings with non-linear, graph-smooth reward structures.

**Notations.** Let $[n]$ be set $\{1, 2, ..., n\}$. For a set or event $\mathcal{E}$, we denote its complement as $\bar{\mathcal{E}}$. Vectors are assumed to be column vectors. $\boldsymbol{e}_i$ is the i-th canonical basis vector in $\mathbb{R}^n$. $\boldsymbol{I}$ is the identity matrix. $\lambda_{\min}(\boldsymbol{A})$ represents the minimum eigenvalue of matrix $\boldsymbol{A}$. $\otimes$ is the Kronecker product. Denote the history of randomness up to (but not including) round $t$ as $\mathcal{F}_t$ and write $\mathbb{P}_t(\cdot) := \mathbb{P}(\cdot | \mathcal{F}_t)$ and $\mathbb{E}_t(\cdot) := \mathbb{E}[\cdot | \mathcal{F}_t]$ for the conditional probability and expectation given $\mathcal{F}_t$. We use $\tilde{\mathcal{O}}$ for big-$O$ notation up to logarithmic factor and $\asymp$ to represent asymptotically equivalence in rate of growth for any two functions.

## 2 PROBLEM FORMULATION

### 2.1 GANG OF BANDITS WITH NON-LINEAR REWARDS

We consider a multi-user contextual bandit problem, often called Gang of Bandits (GOB) Cesa-Bianchi et al. (2013), with $n$ users and a potentially infinite set of arms. We denote the set of users as $\mathcal{U} = \{1, \ldots, n\}$ and the arm set as $\mathcal{D} \subseteq \mathbb{R}^d$, where each arm is represented by a feature vector $\boldsymbol{x} \in \mathcal{D}$. The users are connected by a known undirected graph $G = (\mathcal{U}, \mathcal{E})$, where $\mathcal{E}$ is the set of edges. Let $\boldsymbol{W} \in \mathbb{R}^{n \times n}$ be the matrix of non-negative edge weights $w_{ij}$, and $\boldsymbol{D}$ be the diagonal degree matrix with entries $d_i := \sum_j w_{ij}$. The corresponding graph Laplacian is $\boldsymbol{L} := \boldsymbol{D} - \boldsymbol{W}$.

The learning process unfolds over $T$ rounds. At each round $t \in \{1, \ldots, T\}$, the environment presents a user $u_t \in \mathcal{U}$ (for example, randomly/uniformly pick one) and a finite subset of avail-

able arms $\mathcal{D}_t \subseteq \mathcal{D}$. The learner selects an arm $\boldsymbol{x}_t \in \mathcal{D}_t$ following some decision policy $\pi$ and observes a noisy reward: $y_t = f_{u_t}(\boldsymbol{x}_t) + \epsilon_t$ where $\{f_u : \mathcal{D} \to \mathbb{R}\}_{u=1}^n$ is a collection of unknown reward functions, one for each user. The noise term $\epsilon_t$ is assumed to be conditionally zero-mean and sub-Gaussian with variance proxy $\sigma^2$, given the history of interactions $\mathcal{F}_t$. For the illustrative purpose, we use $f_{1:n} := \{f_u\}_{u=1}^n$ as the collection of the user-level reward functions.

The learner's objective is to minimize cumulative regret. The instantaneous regret incurred at time $t$ is $\Delta_t = f_{u_t}(\boldsymbol{x}_t^*) - f_{u_t}(\boldsymbol{x}_t)$ where $\boldsymbol{x}_t^* = \arg\max_{\boldsymbol{x} \in \mathcal{D}_t} f_{u_t}(\boldsymbol{x})$ and and cumulative regret over $T$ rounds is defined as $\mathcal{R}_T = \sum_{t=1}^T \Delta_t$. A successful algorithm must achieve sub-linear regret, $\mathcal{R}_T/T \to 0$ as $T \to \infty$, ensuring that the average per-round regret vanishes.

## 2.2 A Principled Regularity Model for Graph Homophily

To make learning tractable, we need to impose regularity on the unknown functions $f_{1:n}$. We make two core assumptions. First, we assume that each function $f_u$ is individually well-behaved, belonging to a common Reproducing Kernel Hilbert Space (RKHS), denoted $\mathcal{H}_x$, with a positive semi-definite kernel $K_x : \mathcal{D} \times \mathcal{D} \to \mathbb{R}$. The associated feature map is denoted as $\varphi$ such that $K_x(\boldsymbol{x}, \boldsymbol{x}') = \langle \varphi(\boldsymbol{x}), \varphi(\boldsymbol{x}') \rangle_{\mathbb{R}^d}$. This captures the non-linear structure of rewards with respect to arm features.

Second, we formalize the notion of graph homophily by assuming that users connected by an edge in $G$ have similar reward functions. This *user similarity* is measured by the squared distance between functions in the RKHS, $\|f_i - f_j\|_{\mathcal{H}_x}^2$. Combining these principles, we model the true reward functions as having a small joint penalty that balances graph smoothness with individual function complexity:

$$\mathrm{PEN}(f_{1:n}; \rho) := \underbrace{\frac{1}{2} \sum_{i,j=1}^n w_{ij} \|f_i - f_j\|_{\mathcal{H}_x}^2}_{\mathrm{PEN}_{graph}(f_{1:n})} + \rho \underbrace{\sum_{i=1}^n \|f_i\|_{\mathcal{H}_x}^2}_{\mathrm{PEN}_{ridge}(f_{1:n})} = \sum_{i,j=1}^n [\boldsymbol{L}_\rho]_{ij} \langle f_i, f_j \rangle_{\mathcal{H}_x}, \quad (1)$$

where $\rho > 0$ is a regularization hyperparameter and $\boldsymbol{L}_\rho := \boldsymbol{L} + \rho \boldsymbol{I}$ is the regularized graph Laplacian. This penalty is central to our framework, as it provides a clear, interpretable objective for modeling related, non-linear functions.

## 2.3 From Joint Penalty to a Unified Multi-user Kernel

Our key theoretical insight is that the intuitive, additive penalty in equation 1 is not merely an ad-hoc regularizer. It is, in fact, the squared norm in a single, unified Hilbert space over the user-arm product domain $\mathcal{U} \times \mathcal{D}$. This allows us to reframe the problem from learning $n$ related functions to learning one "lifted" function, $f(\boldsymbol{x}, u) := f_u(\boldsymbol{x})$, in this new space. We show that it is the squared RKHS norm for the product space $\mathcal{H} = \mathcal{H}_G \otimes \mathcal{H}_x$ where $\mathcal{H}_G$ is the RKHS with kernel $K_G(u, u') = [\boldsymbol{L}_\rho^{-1}]_{u,u'}$ in the following theorem.

**Theorem 2.1** (Multi-user Kernel). *Let $\mathcal{H}_x$ be an RKHS of functions on $\mathcal{D}$ with kernel $K_x$. The vector space of function collections $\mathcal{H} := \{(f_1, \ldots, f_n) : f_u \in \mathcal{H}_x, \forall u \in \mathcal{U}\}$ equipped with the inner product*

$$\langle f, g \rangle_{\mathcal{H}} := \sum_{i,j=1}^n [\boldsymbol{L}_\rho]_{ij} \langle f_i, g_j \rangle_{\mathcal{H}_x}$$

*is a Reproducing Kernel Hilbert Space of functions on $\mathcal{U} \times \mathcal{D}$. The associated squared RKHS norm is precisely the penalty in equation 1, and its reproducing kernel $K : (\mathcal{D} \times \mathcal{U})^2 \to \mathbb{R}$ is given by:*

$$K((\boldsymbol{x}, u), (\boldsymbol{x}', u')) = [\boldsymbol{L}_\rho^{-1}]_{u,u'} K_x(\boldsymbol{x}, \boldsymbol{x}'). \quad (2)$$

This result is powerful: it provides a direct, canonical construction for a *multi-user kernel* that fuses graph and feature information. The kernel $K_x$ captures similarity between arms, while the matrix $\boldsymbol{L}_\rho^{-1}$ (the graph Green's function) captures similarity between users, with $[\boldsymbol{L}_\rho^{-1}]_{u,u'}$ measuring the strength of connection between users $u$ and $u'$ through all paths in the graph. See Appendix A for more background.

This unification allows us to represent the lifted reward function $f(\boldsymbol{x}, u)$ via a feature map $\phi(\boldsymbol{x}, u)$ such that $f(\boldsymbol{x}, u) = \langle \boldsymbol{\theta}, \phi(\boldsymbol{x}, u) \rangle$ for some (potentially infinite-dimensional) parameter $\boldsymbol{\theta}$, and $K((\boldsymbol{x}, u), (\boldsymbol{x}', u')) = \langle \phi(\boldsymbol{x}, u), \phi(\boldsymbol{x}', u') \rangle$. Formally, for a context-user pair $(\boldsymbol{x}, u) \in \mathcal{D} \times \mathcal{U}$, the feature map $\phi$ is defined as $\phi(\boldsymbol{x}, u) := \boldsymbol{L}_\rho^{-1/2} \boldsymbol{e}_u \otimes \varphi(\boldsymbol{x})$. The problem is now cast as learning a single function in the *multi-user RKHS* $\mathcal{H}$. This insight paves the way for a principled algorithmic approach based on Gaussian processes, which we detail next.

## 3 Laplacian Kernelized Bandit Algorithms

The identification of the *multi-user RKHS* with its explicit kernel $K$ provides a powerful, unified framework for the GOB problem. It allows us to model the entire system—across all users and arms—with a single Gaussian Process (GP), sidestepping the complexity of managing $n$ separate but correlated models.

### 3.1 A Gaussian Process Perspective

We propose algorithms based on the Gaussian process (GP), motivated by the kernelized bandit literature Chowdhury & Gopalan (2017). Our Bayesian modeling is only assumed for derivation of our estimators and it is not necessarily the true model. We place a GP prior over the unknown lifted reward function $f : \mathcal{D} \times \mathcal{U} \to \mathbb{R}$, denoted as

$$[f_1(\cdot), \ldots, f_n(\cdot)] \sim \mathcal{GP}(0, K(\cdot, \cdot)).$$

where $K$ is the multi-user kernel defined in equation 2. For any finite set of user-arm pairs $\{(\boldsymbol{x}_i, u_i)\}_{i=1}^t$, This proir implies that $\boldsymbol{f}_t := [f_{u_1}(\boldsymbol{x}_1), \cdots, f_{u_t}(\boldsymbol{x}_t)]^\top \sim \mathcal{N}(\boldsymbol{0}, \boldsymbol{K}_t)$ where $\boldsymbol{K}_t \in \mathbb{R}^{t \times t}$ with entries $[\boldsymbol{K}_t]_{ij} = K((\boldsymbol{x}_i, u_i), (\boldsymbol{x}_j, u_j))$ is the kernel matrix.

At round $t$, given user $u_t$ and selected arm $\boldsymbol{x}_t$, the Bayesian model assume a reward model $y_t = f(\boldsymbol{x}_t, u_t) + \epsilon_t$ where $\epsilon_t \sim \mathcal{N}(0, \lambda)$ is the noise. Therefore, conditioned on the history $\mathcal{F}_t$, the posterior distribution for $f_u(\boldsymbol{x})$ is $\mathcal{N}(\mu_{u,t-1}(\boldsymbol{x}), \sigma_{u,t-1}^2(\boldsymbol{x}))$, with the posterior mean and variance:

$$\begin{aligned} \mu_{u,t}(\boldsymbol{x}) &= \boldsymbol{k}_t(\boldsymbol{x}, u)^\top (\boldsymbol{K}_t + \lambda \boldsymbol{I}_t)^{-1} \boldsymbol{y}_t \\ \sigma_{u,t}^2(\boldsymbol{x}) &= K((\boldsymbol{x}, u), (\boldsymbol{x}, u)) - \boldsymbol{k}_t(\boldsymbol{x}, u)^\top (\boldsymbol{K}_t + \lambda \boldsymbol{I}_t)^{-1} \boldsymbol{k}_t(\boldsymbol{x}, u). \end{aligned} \tag{3}$$

Here $\boldsymbol{k}_t(\boldsymbol{x}, u) := [K((\boldsymbol{x}_1, u_1), (\boldsymbol{x}, u)), \ldots, K((\boldsymbol{x}_t, u_t), (\boldsymbol{x}, u))]^\top \in \mathbb{R}^t$ is the kernel vector between past selected user-action pairs $\{(\boldsymbol{x}_s, u_s)\}_{s=1}^t$ and new pair $(\boldsymbol{x}, u)$, and $\boldsymbol{y}_t = [y_1, \ldots, y_t]^\top \in \mathbb{R}^t$ is the observed reward.

**Remark 1.** *When $\{(u_t, \boldsymbol{x}_t)\}_{t=1}^T$ is a fixed (deterministic) sequence, under this model we have $\boldsymbol{y}_t \mid \boldsymbol{f}_t \sim N(\boldsymbol{f}_t, \lambda \boldsymbol{I}_t)$ and $\boldsymbol{f}_t \sim N(\boldsymbol{0}, \boldsymbol{K}_t)$. Then, the mutual information between $\boldsymbol{y}_t$ and $\boldsymbol{f}_t$ is given by: $I(\boldsymbol{y}_t; \boldsymbol{f}_t) = \frac{1}{2} \log \det(\boldsymbol{I}_t + \lambda^{-1} \boldsymbol{K}_t)$, which is often referred as the information gain of the Bayesian model (Srinivas et al., 2009, Section 2.1). For convenience, we write*

$$\gamma_t := \log \det(\boldsymbol{I}_t + \lambda^{-1} \boldsymbol{K}_t), \tag{4}$$

*and refer to it as the information gain at round $t$, although it is twice what is usually called the information gain in the literature. Moreover, $\gamma_t$ in our notation depends on the sequence, although in the literature, this symbols is often used for the maximum information gain over all sequence $\{(u_t, \boldsymbol{x}_t)\}_{t=1}^T$ of length $T$.*

**Connection to Regularized Regression.** It is worth noting that the GP posterior mean estimator in equation 3 is equivalent to the solution of an offline Kernel Laplacian Regularized Regression (KLRR) problem. Specifically, the function $f \in \mathcal{H}$ that minimizes the regularized least-squares objective

$$\min_{f \in \mathcal{H}} \sum_{s=1}^t (f(\boldsymbol{x}_s, u_s) - y_s)^2 + \lambda \|f\|_{\mathcal{H}}^2 \tag{5}$$

is precisely the posterior mean function $\mu_{t-1}(\boldsymbol{x}, u)$. This equivalence confirms that our online, GP-based algorithm is deeply connected to the batch learning principle of minimizing prediction error regularized by our proposed multi-user RKHS norm from equation 1.

## 3.2 Decision Strategies: UCB and Thompson Sampling

With these posterior estimates, we can design bandit algorithms that effectively balance exploration and exploitation. We propose two algorithms based on common and powerful heuristics: Upper Confidence Bound (UCB) and Thompson Sampling (TS). The complete procedures are described in Appendix E.1

**Laplacian Kernelized GP-UCB (`LK-GP-UCB`).** Following the principle of "optimism in the face of uncertainty," our UCB algorithm selects the arm with the highest optimistic estimate of the reward. At round $t$, upon observing user $u_t$ and arm set $\mathcal{D}_t$, it chooses:

$$\boldsymbol{x}_t = \arg\max_{\boldsymbol{x} \in \mathcal{D}_t} \Big( \mu_{u_t, t-1}(\boldsymbol{x}) + \beta_t \sigma_{u_t, t-1}(\boldsymbol{x}) \Big), \tag{6}$$

where $\beta_t$ is the hyperparameter that ensures the appropriate scale of exploration via confidence width $\sigma_{u_t, t-1}(\boldsymbol{x})$. Our theoretical analysis provides an explicit form for $\beta_t$ in Theorem 4.2 to guarantee low regret, though in practice it is often treated as a tunable hyperparameter.

**Laplacian Kernelized GP-TS (`LK-GP-TS`).** Thompson Sampling Thompson (1933); Russo et al. (2018) operates on the principle of "probability matching." At each round, it draws a random function from the posterior distribution and acts greedily with respect to this sample. A practical way to implement this is to select the arm that maximizes a sample from the posterior predictive distribution for the reward:

$$\boldsymbol{x}_t = \arg\max_{\boldsymbol{x} \in \mathcal{D}_t} \Big( \mu_{u_t, t-1}(\boldsymbol{x}) + \nu_t z_t(\boldsymbol{x}) \sigma_{u_t, t-1}(\boldsymbol{x}) \Big), \tag{7}$$

where $\nu_t$ is the scale hyparameter for exploration and $z_t(\boldsymbol{x}) \sim \mathcal{N}(0,1)$ is the Gaussian perturbation. Aligned with common Thompson Sampling literature, our decision strategy in equation 7 can be separated into two steps: sampling $\widetilde{\mu}_t(\boldsymbol{x})$ from $\mathcal{N}(\mu_{u_t, t-1}(\boldsymbol{x}), \nu_t^2 \sigma_{u_t, t-1}^2(\boldsymbol{x}))$ for all $\boldsymbol{x} \in \mathcal{D}_t$ and choosing an arm by $\boldsymbol{x}_t = \arg\max_{\boldsymbol{x} \in \mathcal{D}_t} \widetilde{\mu}_t(\boldsymbol{x})$. Similarly to the UCB algorithm, we also use the explicit theoretical choice for $\nu_t$ in Theorem 4.3, while it is a tuning hyperparameter in a real application.

## 3.3 Practical Implementation

A naive implementation of the posterior updates in equation 3 is computationally expensive, requiring an $\mathcal{O}(t^3)$ matrix inversion at each step. To ensure practical scalability, we can use recursive formulas to update the posterior mean and variance in $\mathcal{O}(t^2)$ or, for a fixed grid of points, even more efficiently. Specifically, we can maintain and update the inverse matrix $(\boldsymbol{K}_t + \lambda \boldsymbol{I}_t)^{-1}$ or use the following recursive updates for the posterior estimators Chowdhury & Gopalan (2017):

$$\mu_{u,t}(\boldsymbol{x}) = \mu_{u,t-1}(\boldsymbol{x}) + \frac{q_{t-1}((\boldsymbol{x}, u), (\boldsymbol{x}_t, u_t))}{\lambda + \sigma_{u_t, t-1}^2(\boldsymbol{x}_t)}(y_t - \mu_{u_t, t-1}(\boldsymbol{x}_t))$$

$$q_t((\boldsymbol{x}, u), (\boldsymbol{x}', u')) = q_{t-1}((\boldsymbol{x}, u), (\boldsymbol{x}', u')) - \frac{q_{t-1}((\boldsymbol{x}, u), (\boldsymbol{x}_t, u_t)) q_{t-1}((\boldsymbol{x}_t, u_t), (\boldsymbol{x}', u'))}{\lambda + \sigma_{u_t, t-1}^2(\boldsymbol{x}_t)} \tag{8}$$

$$\sigma_{u,t}^2(\boldsymbol{x}) = \sigma_{u,t-1}^2(\boldsymbol{x}) - \frac{q_{t-1}^2((\boldsymbol{x}, u), (\boldsymbol{x}_t, u_t))}{\lambda + \sigma_{u_t, t-1}^2(\boldsymbol{x}_t)}.$$

where $q_t((\boldsymbol{x}, u), (\boldsymbol{x}', u'))$ is the estimated posterior covariance at round $t$. We explain how to obtain the updates in Appendix E.2. A hybrid approach that uses exact inversion for small $t$ and switches to recursive updates for larger $t$ can balance numerical stability and computational efficiency. Further details on our implementation are provided in Appendix F.4.

## 4 Regret Analysis

We now provide theoretical guarantees for our proposed algorithms. Our analysis is built upon a high-probability confidence bound for our GP posterior estimates, which in turn leads to sub-linear regret bounds for both `LK-GP-UCB` and `LK-GP-TS`.

## 4.1 ASSUMPTIONS

Our results rely on the following standard assumptions.

**Assumption 1** (Sub-Gaussian Noise). *The noise process $\{\epsilon_t\}_{t=1}^T$ is a $\mathcal{F}_t$-measurable stochastic process and is conditionally sub-Gaussian with constant $\sigma^2$.*

**Assumption 2** (Bounded Base Kernel). *The base arm kernel $K_x(\cdot, \cdot)$ is positive semi-definite and its diagonal is uniformly bounded: $\sup_{\boldsymbol{x} \in \mathcal{D}} K_x(\boldsymbol{x}, \boldsymbol{x}) \leq \alpha^2$ for some $\alpha > 0$.*

**Assumption 3** (Bounded Multi-User RKHS Norm). *The true lifted reward function $f$ has a bounded norm in the* multi-user RKHS $\mathcal{H}$*: $\|f\|_{\mathcal{H}}^2 = PEN(f_{1:n}; \rho) \leq B_\rho^2$ for some constant $B_\rho > 0$.*

Assumption 1 is common assumption in bandit literature. Assumption 2 and 3 indirectly align with the regularity assumptions in kernelized bandit and graph smoothness literatures Belkin et al. (2006); Kocák et al. (2020). These assumptions imply that the rewards and the *multi-user kernel* are bounded. Formally, we have

$$\sup_{(\boldsymbol{x}, u) \in \mathcal{D} \times \mathcal{U}} K((\boldsymbol{x}, u), (\boldsymbol{x}, u)) \leq K_{\max} := \alpha^2 \cdot \max_{u \in \mathcal{U}}[\boldsymbol{L}_\rho^{-1}]_{u, u}.$$

## 4.2 HIGH PROBABILITY CONFIDENCE BOUND

The core of our regret analysis is the confidence bound that relates the true function $f$ to our posterior mean estimator $\mu_t$. This result quantifies the model's uncertainty and justifies the exploration strategy of the UCB algorithm.

**Theorem 4.1** (Confidence Bound). *Suppose Assumptions 1, 2, and 3 hold. Let $\{(\boldsymbol{x}_t, u_t)\}_{t=1}^\infty$ be the $\mathcal{F}_{t-1}$-measurable discrete time stochastic process. Then, using the posterior estimators $\mu_{u,t}(\boldsymbol{x})$ and $\sigma_{u,t}(\boldsymbol{x})$ in equation 3 yields to a high probability upper bound: for any $\delta \in (0, 1)$, with probability at least $1 - \delta$, for all $t \geq 1$ and all $(\boldsymbol{x}, u) \in \mathcal{D} \times \mathcal{U}$:*

$$|\mu_{u,t}(\boldsymbol{x}) - f(\boldsymbol{x}, u)| \leq \beta_t \cdot \sigma_{u,t}(\boldsymbol{x}) \tag{9}$$

*where the confidence parameter $\beta_t$ is given by*

$$\beta_t := B_\rho + \sqrt{\frac{\sigma^2}{\lambda}\left(2\log\frac{1}{\delta} + \log\det(\boldsymbol{I}_t + \lambda^{-1}\boldsymbol{K}_t)\right)}. \tag{10}$$

This confidence bound follows a structure similar to those in the kernelized bandit literature Chowdhury & Gopalan (2017); Valko et al. (2013); Dubey et al. (2020), but our analysis offers two key distinctions. First, our proof does not require the constraint $\lambda \geq 1$ found in some prior work. More significantly, we retain the term $\log\det(\boldsymbol{I}_t + \lambda^{-1}\boldsymbol{K}_t)$ directly within our confidence width $\beta_t$. This contrasts with classical approaches that often proceed by further bounding this term using information-theoretic quantities, which can result in looser bounds. By keeping the exact term, we set the stage for a tighter, data-dependent analysis via the effective dimension.

## 4.3 REGRET BOUNDS VIA EFFECTIVE DIMENSION

To obtain concrete regret rates, we characterize the growth of the $\log\det$ term using the notion of an *effective dimension*.

**Definition 4.1** (Effective Dimension). *The effective dimension $\tilde{d}$ of the learning problem, given the sequence of actions up to time $T$, is defined as:*

$$\tilde{d} := \frac{\log\det(\boldsymbol{I}_T + \boldsymbol{K}_T/\lambda)}{\log(1 + TK_{\max}/\lambda)}. \tag{11}$$

This quantity, inspired by recent work in kernel methods and overparameterized models Wu & Amini (2024); Bietti & Mairal (2019); Yang & Wang (2020), measures the intrinsic complexity of the learning problem. It can be interpreted as the ratio of the sum of log-eigenvalues of the matrix $\boldsymbol{I}_T + \boldsymbol{K}_T/\lambda$ to a bound on the maximum possible log-eigenvalue ($TK_{\max}$ is an upper bound on the largest eigenvalue of $\boldsymbol{K}_T$). As such, it serves as a robust, graph-dependent measure of the matrix's rank, capturing the "dimensionality" of the function space actually explored by the algorithm.

Using the confidence bound in Theorem 4.1 and $\tilde{d}$ in Definition 4.1, we provide the regret upper bound for `LK-GP-UCB` and `LK-GP-TS` as follow.

**Theorem 4.2** (Regret Bound of `LK-GP-UCB`). *Suppose Assumptions1, 2 and 3 hold, with no assumption on the number of arms. By setting the exploration parameter $\beta_t$ in `LK-GP-UCB` to $\beta_t$ from Theorem 4.1, the cumulative regret is bounded with high probability as:*

$$\mathcal{R}_T = \mathcal{O}(\tilde{d}\log(T)\sqrt{T}) = \tilde{\mathcal{O}}(\tilde{d}\sqrt{T})$$

**Theorem 4.3** (Regret Bound of `LK-GP-TS`). *Suppose Assumptions1, 2 and 3 hold, and the decision sets $\mathcal{D}_t$ are uniformly finite. By setting the exploration parameter $\nu_t$ in `LK-GP-TS` to $\beta_t$ from Theorem 4.1, the cumulative regret is bounded with high probability as:*

$$\mathcal{R}_T = \mathcal{O}(\tilde{d}\log(T)^{3/2}\sqrt{T}) = \tilde{\mathcal{O}}(\tilde{d}\sqrt{T})$$

These bounds demonstrate the efficiency of our approach. The regret scales not with the number of users $n$ or the ambient feature dimension, but with the effective dimension $\tilde{d}$. For problems where the graph and kernel structure lead to a rapid spectral decay, $\tilde{d}$ can be significantly smaller, resulting in substantial gains in sample efficiency.

In the notation of Remark 1, the effective dimension $\tilde{d}$ scales as: $\tilde{d} = \gamma_T / \log(1 + TK_{\max}/\lambda) \asymp \frac{\gamma_T}{\log T}$ where the approximation assumes $\lambda = \Theta(1)$. The interpretation of $\tilde{d}$ as a dimension is evident in the linear setting ($n = 1$ with linear kernel on $\mathbb{R}^d$), where $\gamma_T = \mathcal{O}(d\log T)$ (Srinivas et al., 2009, Theorem 5), yielding $\tilde{d} = \mathcal{O}(d)$. This example demonstrates that our bound $\tilde{\mathcal{O}}(\tilde{d}\sqrt{T})$ is tight up to logarithmic factors for infinite action spaces, matching the minimax optimal rate $\tilde{\mathcal{O}}(d\sqrt{T})$ for linear bandits Dani et al. (2008).

For uniformly finite action spaces ($|\mathcal{D}_t| \leq M$ for all $t$), it is possible to achieve a tighter regret bound of $\tilde{\mathcal{O}}(\sqrt{\tilde{d}T})$ using algorithms such as SupKernelUCB Valko et al. (2013). This improvement relies on scaling the exploration parameter as $\beta_t \propto 1/\sqrt{\lambda}$ rather than using equation 10, effectively removing a factor of $\sqrt{\gamma_T}$. Since our primary contribution is the construction of the unified multi-user kernel, such algorithmic refinements from the kernel bandit literature are directly applicable to our framework.

## 4.4 SPECTRAL ANALYSIS OF THE MULTI-USER KERNEL

To interpret the effective dimension $\tilde{d}$, we analyze the spectrum of the multi-user kernel $K$. By Theorem 2.1, $K = K_G \otimes K_x$, the tensor product of the user kernel $K_G$ associated with matrix $\boldsymbol{K}_G = \boldsymbol{L}_\rho^{-1}$ and the arm kernel $K_x$. Consequently, the eigenvalues of the integral operator associated with $K$ are the pairwise products of the marginal eigenvalues. Let $\{\lambda_i^G\}_{i=1}^n$ be the eigenvalues of $\boldsymbol{L}_\rho^{-1}$ and $\{\nu_j^x\}_{j=1}^\infty$ be the eigenvalues of $K_x$. The operator eigenvalues for $K$ are then $\{\mu_{ij} = \lambda_i^G \nu_j^x\}_{i,j}$. The eigenvalues of the normalized matrix $\boldsymbol{K}_T/T$ approximate these operator eigenvalues[1].

In particular, we obtain the following approximate upper bound on the information gain $\gamma_T$:

$$\gamma_T = \log\det\left(\boldsymbol{I} + \lambda^{-1}\boldsymbol{K}_T\right) \lesssim\approx \sum_{i=1}^n \sum_{j=1}^\infty \log\left(1 + \frac{T}{\lambda}\lambda_i^G \nu_j^x\right) = \sum_{i=1}^n \Psi\left(\frac{T\lambda_i^G}{\lambda}\right), \quad (12)$$

where $\Psi(s) := \sum_{j=1}^\infty \log\left(1 + s\nu_j^x\right)$ represents the information gain of a single-user problem with effective signal strength $s$. We know that $\Psi(s)$ is concave and sublinear; e.g., for the squared exponential kernel on $\mathbb{R}^d$, $\Psi(s) \lesssim (\log s)^{d+1}$ (Srinivas et al., 2009, Theorem 5)), hence as a function of $T$, $\gamma_T$ grows slowly in $T$. What is interesing then is the dependence on $n$.

While informative, the bound in equation 12 can be conservative for finite $T$ (see Figure 1). A sharper bound in a similar vein can be obtained by considering a *regular design*: assume we observe each user exactly $m := T/n$ times, choosing the same set of actions $\{\boldsymbol{x}_1, \cdots, \boldsymbol{x}_m\}$ for all users. By permuting round indices such that all observations for user 1 appear first, followed by user 2, etc., the eigenvalues of $\boldsymbol{K}_T$ remain invariant. Under this setup, $\boldsymbol{K}_T = \boldsymbol{K}_G \otimes \boldsymbol{K}_x^{\text{base}}$, where $\otimes$ is the matrix Kronecker product and $\boldsymbol{K}_x^{\text{base}}$ is the $m \times m$ kernel matrix evaluated on the common action set. Let $\{\hat{\nu}_j^x\}_{j=1}^m$ be the eigenvalues of $\boldsymbol{K}_x^{\text{base}}/m$. The normalization by $m$ ensures that $\hat{\nu}_j^x$ stabilize around the population eigenvalues $\nu_j^x$ for large $m$.

---

[1] This holds asymptotically as $T \to \infty$ under i.i.d. sampling Koltchinskii & Giné (2000); results from (Srinivas et al., 2009, Theorem 5) suggest a similar approximation holds for worst-case sequences.

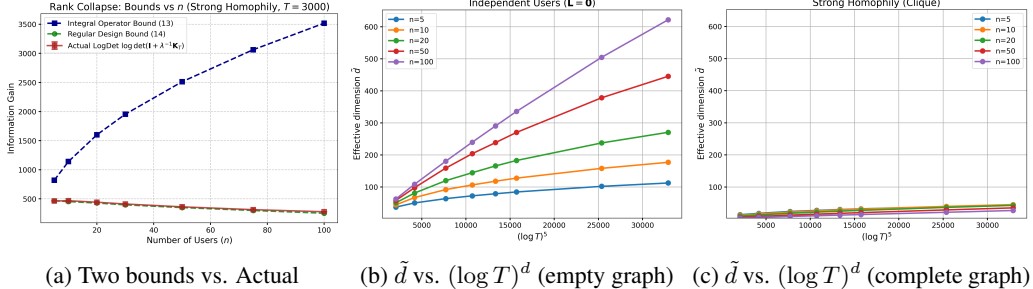

(a) Two bounds vs. Actual    (b) $\tilde{d}$ vs. $(\log T)^d$ (empty graph)   (c) $\tilde{d}$ vs. $(\log T)^d$ (complete graph)

Figure 1: Rank Collapse: (a) Comparing the growth of the actual information gain $\gamma_T$ vs $n$ in i.i.d. design (red) versus the two bounds equation 12 (blue; crude) and equation 13 (green; nearly exact) in a complete graph. The kernel is $\exp\left(-\|x-y\|^2/2\right)$, $u_i \sim \text{Unif}([n])$ and $x_i \sim \text{Unif}[0,1]^d$ where $d = 5$. Panels (b) and (c) show the growth of $\tilde{d}$ vs. $(\log T)^d$ under empty and complete graphs, respectively. Note that under the complete graph, $\tilde{d}$ slightly decreases as $n$ increases.

Consequently, the eigenvalues of $\boldsymbol{K}_T/T$ are given by $\lambda_i^G \hat{\nu}_j^x/n$, yielding the exact expression:

$$\gamma_T = \sum_{i=1}^n \sum_{j=1}^m \log\left(1 + \frac{T}{n\lambda}\lambda_i^G \hat{\nu}_j^x\right) = \sum_{i=1}^n \hat{\Psi}\left(\frac{T}{n\lambda}\lambda_i^G\right), \tag{13}$$

where $\hat{\Psi}(s) := \sum_{j=1}^m \log\left(1 + s\hat{\nu}_j^x\right)$ represents the "empirical" information gain of a single-user problem with common actions. For large enough $m = T/n$, we have $\hat{\nu}_j^x \approx \nu_j^x$ and $\hat{\Psi}(s) \approx \Psi(s)$. Expression equation 13 is exact for regular designs and, as shown in Figure 1, provides a sharp approximation for the i.i.d. sampling case. We use equation 13 to analyze $\tilde{d}$ across graph structures.

**Case 1: Independent Users (Worst Case).** If $\boldsymbol{L} = \boldsymbol{0}$, then $\mathbf{K}_G = \rho^{-1}\boldsymbol{I}$, and $\lambda_i^G = \rho^{-1}$ for all $i \in [n]$. The gain sums linearly: $\gamma_T^{\text{indep}} = \sum_{i=1}^n \hat{\Psi}\left(\frac{T}{n\rho\lambda}\right) = n \cdot \hat{\Psi}\left(\frac{T}{n\rho\lambda}\right)$. Thus, the effective dimension scales as $n$ times the single-user effective dimension. For example, with an SE kernel, $\tilde{d} = \mathcal{O}(n\left(\log(T/n)\right)^d)$, which remains sublinear in $T$.

**Case 2: Strong Homophily (Complete Graph).** To isolate the effect of an extremely dense user graph under a homophilous prior, consider a complete graph with edge weights $w_{ij} = 1$. The Laplacian eigenvalues are $0$ (multiplicity 1) and $n$ (multiplicity $n-1$). The kernel eigenvalues invert this structure, with $\lambda_1^G = 1/\rho$ and $\lambda_i^G = 1/(n+\rho)$ for $i \geq 2$.. For large $n$, this yields a nearly rank-1 matrix. Substituting into equation 13 provides a "Head + Tail" decomposition:

$$\gamma_T^{\text{clique}} = \hat{\Psi}\left(\frac{T}{n\rho\lambda}\right) + (n-1)\hat{\Psi}\left(\frac{T}{n(n+\rho)\lambda}\right). \tag{14}$$

This leads to the following consequence:

**Proposition 4.1.** *Consider the regime where $T \leq Cn$ for some constant $C$. Then, under a regular design:* $\gamma_T^{\text{clique}} \lesssim \frac{C}{\lambda}\left(\frac{1}{\rho} + 1\right) = \mathcal{O}(1)$.

*Proof.* Using $\log(1+x) \leq x$ for $x \geq 0$, we have $\hat{\Psi}(s) \leq s\left(\sum_{j=1}^m \hat{\nu}_j^x\right)$. Then, for $T \leq Cn$,

$$(n-1)\hat{\Psi}\left(\frac{T}{n(n+\rho)\lambda}\right) \leq n\hat{\Psi}\left(\frac{C}{(n+\rho)\lambda}\right) \leq n \cdot \frac{C}{(n+\rho)\lambda}\sum_{j=1}^m \hat{\nu}_j^x \lesssim \frac{C}{\lambda},$$

since $\sum_{j=1}^m \hat{\nu}_j^x = \mathcal{O}(\sum_{j=1}^\infty \nu_j^x) = \mathcal{O}(1)^2$. Similarly, for the first term, $\hat{\Psi}\left(\frac{T}{n\rho\lambda}\right) \lesssim \frac{C}{\rho\lambda}$. $\square$

---

[2]This bound holds for any kernel whose integral operator is trace class. For a uniformly bounded kernel as in Assumption 2, we have the more straightforward bound $\sum_{j=1}^m \hat{\nu}_j^x = \text{tr}\left(\boldsymbol{K}_x^{\text{base}}\right)/m \leq \alpha^2$.

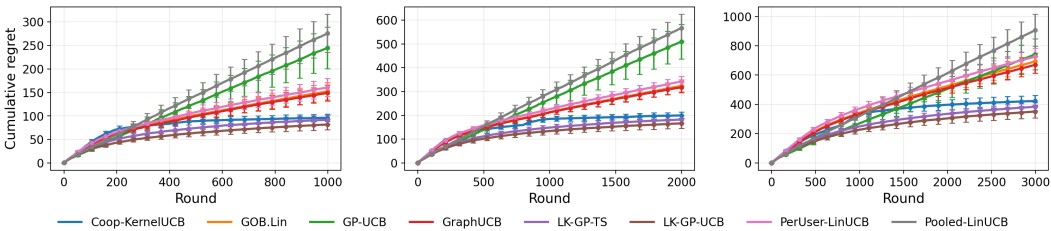

Figure 2: Cumulative Regret under *Linear-GOB* regime. From left to right are tasks of easy level, medium level, to hard level.

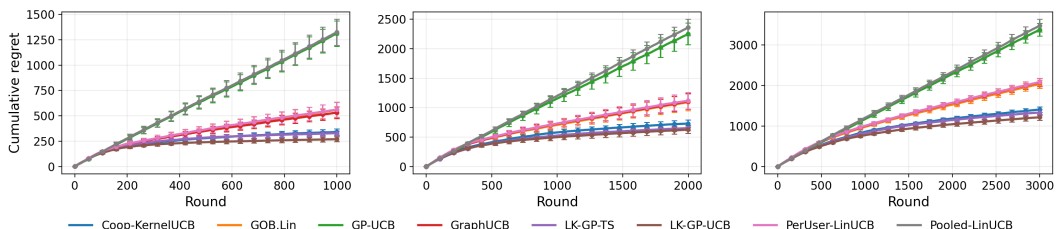

Figure 3: Cumulative Regret under *Laplacian–Kernel* regime using GP draw. From left to right are tasks of easy level, medium level, to hard level.

This result is significant: in the regime $T \leq Cn$, the information gain grows with neither $n$ nor $T$, implying $\tilde{d} = \mathcal{O}(1/\log T)$ (slowly decreasing). This behavior has no counterpart in the single-user setting and confirms that under strong homophily, regret is independent of $n$. These theoretical findings are corroborated by our empirical plots in Figure 1.

**Generalization to Clusters.** If the graph contains $k$ disjoint clusters with high internal connectivity, $\mathbf{K}_G$ will have $k$ eigenvalues of magnitude $\mathcal{O}(1)$ and $n-k$ of magnitude $\mathcal{O}(1/n)$. A similar argument implies that $\tilde{d} = \mathcal{O}(k/\log T)$ when $T \leq Cn$. Thus, $\tilde{d}$ essentially counts the number of significant eigenvalues of the normalized kernel $\mathbf{K}_G$, serving as a soft proxy for the number of distinct user clusters.

**Comparison with Independent Bandits.** It is instructive to compare this with independent learners that share no information. Since each user generates $T/n$ observations on average, the regret for learning each function is at best $\sqrt{T/n}$, yielding an overall regret of $\sum_{u=1}^{n} \sqrt{T/n} = \sqrt{nT}$. In the worst case (Case 1), our bound $\tilde{d}\sqrt{T}$ scales as $n\sqrt{T}$ (ignoring log factors), which is a factor of $\sqrt{n}$ looser than the independent baseline. However, had we assumed a uniformly finite action space, we could achieve a regret bound of $\sqrt{\tilde{d}T} \asymp \sqrt{nT}$, matching the optimal independent rate.

The advantage of our approach becomes evident under strong homophily. For independent learners in the regime $T \asymp n$, the regret scales as $\sqrt{nT} \asymp T$, meaning no learning occurs. In contrast, we showed that our Laplacian Kernelized Bandit achieves regret of $\mathcal{O}(\sqrt{T})$ in this regime (up to log factors). A similar improvement holds when there are $k = \mathcal{O}(1)$ strong clusters.

## 5 EXPERIMENTS

We evaluate Laplacian Kernelized bandit algorithms, `LK-GP-UCB` and `LK-GP-TS` on several synthetic data environments that capture user–user homophily on a known graph while varying reward structure (linear vs. nonlinear) and problem difficulty. Baseline algorithms include `GraphUCB`Yang et al. (2020), `GoB.Lin`Cesa-Bianchi et al. (2013), `COOP-KernelUCB`Dubey et al. (2020), `GP-UCB`Chowdhury & Gopalan (2017), `Pooled LinUCB` and `Per-User LinUCB`. Full implementation details are Provided in Appendix F.

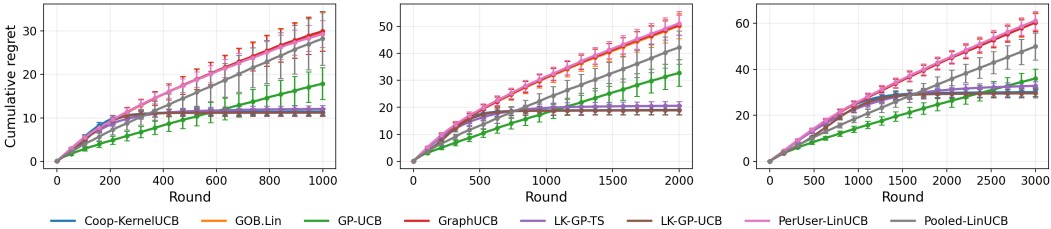

Figure 4: Cumulative Regret under *Laplacian–Kernel* regime using representer draw. From left to right are tasks of easy level, medium level, to hard level.

**Environments.** We draw a context pool $\mathcal{D}$ by sampling from $\mathcal{N}(\mathbf{0}, \boldsymbol{I}_d)$ first and then normalize the context vectors. At round $t$ we present $\mathcal{D}_t$ by sampling $M_t$ distinct items from $\mathcal{D}$ without replacement. We generate the user graphs by Erdős–Rényi (ER) random graph model or Radial basis function(RBF) random graph model. After giving the generated graph, we consider one linear regime and two kernelized(nonlinear) regimes for synthetic data simulation. First synthetic data environment is called *Linear–GOB*. We consider simulating the true graph graph-smooth user parameters $\boldsymbol{\Theta} = (\boldsymbol{I} + \eta \boldsymbol{L})^{-1} \boldsymbol{\Theta}_0$, which enforce graph homophily on the random initial parameters $\boldsymbol{\Theta}_0 \in \mathbb{R}^{n \times d}$ Yang et al. (2020). The homophily strength is controlled by $\eta$ in *Linear–GOB* regime. We also generate the true reward functions by simulating *multi-user kernel*, which is called the *Laplacian–Kernel* regime. We first use *Squared Exponential* as our base kernel $K_x$ over arms $\mathcal{U}$ and construct the *multi-user kernel* using equation 2. Next, we design two choices to generate $f$, including a GP draw and a representer draw. We leave all the details for data simulation in Appendix F.1.

**Task Design.** Our experiment has following design of the bandit tasks for a general comparison. In these tasks, the noise of reward is set as $\sigma = 0.1$ and the number of users is $n = 20$. The simplest level task is a 10-arm bandit problem ($m = 10$) with 50% viewability ($M_t = 5$) at each round for all users, under $T = 1000$ interaction rounds. Medium level task is a 20-arm bandit problem ($m = 20$) with 25% viewability ($M_t = 5$) at each round for all users, under $T = 3000$ interaction rounds. The hard task is a 50-arm bandit problem ($m = 50$) with 10% viewability ($M_t = 5$) at each round for all users, under $T = 5000$ interaction rounds. In our figures (2, 3 and 4), from left to right are tasks of easy level, medium level, to hard level.

**Algorithms Configurations.** Our proposals `LK-GP-UCB` and `LK-GP-TS` are given in Algorithm 1 and Algorithm 2 in Appendix E.1. We implement the hybrid updates using practical recursive update in equation 8 and exact update in equation 3 with Cholesky decomposition. Details are in Appendix F.4. Hyperparameters $\nu$ and $\beta$ are tuned. For `Coop-KernelUCB`, we initially set five choices of similarity kernel $K_z$ and conduct an experiment (Figure in Appendix) to verify that the inverse Laplacian $\boldsymbol{L}_\rho^{-1}$ is the optimal choice while the empirical maximum mean discrepancy method is close to the best choice. In the experiment, $K_z$ is set as the empirical MMD method to learn the similarity kernel $K_z$ unless otherwise stated. The classical baselines for GOB problem, `GoB.Lin`, `GraphUCB`, and all the remaining baselines, `Pooled LinUCB`, `Per-User LinUCB` and `GP-UCB`, are all `UCB`-based algorithms. We also tune their hyperparameter for the confidence bound. The regularization parameter $\lambda$ is is designed as a scheduling $\lambda_t = \lambda_{\text{base}} \cdot S_{\text{spec}} \cdot \frac{T}{T+t}$ where $S_{\text{spec}}$ is the ratio of the smallest non-zero eigenvalue to the max eigenvalue and $\lambda_{\text{base}}$ is tuned. Appendix F.5 discusses hyperparameter tuning. All methods run in a centralized, no-delay setting.

**Main Findings.** Our proposals `LK-GP-UCB` and `LK-GP-TS` have robust performance in all the 9 data environments. In the *Linear-GOB* regime, which is the preferred setting for linear bandit algorithms, our proposals can beat the most baselines with clear gaps. In the *Laplacian-Kernel* regime, our proposals are consistently the best choices. For the GP draw setting, our proposals are always the top algorithms in our experiment. For setting using representer draw, `LK-GP-UCB` and `LK-GP-TS` are sublinear while most baselines are hard to achieve sublinear regret. We believe our proposed algorithms can clearly outperform others in a long-term manner due to the achievement of the clear sublinear regret. Lastly, even though we conduct an empirical study on the choice for `Coop-KernelUCB` and pick a best one in the comparison, leading to the top performances(close to our proposal) of `Coop-KernelUCB`, our `LK-GP-UCB` are consistently better than `Coop-KernelUCB`.

ACKNOWLEDGMENTS

This material is based upon work supported by the National Science Foundation under Award No. 1945667.

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

## A  RELATED WORK

**Gaussian Processes on Graphs**  Our kernel construction builds upon foundational work in graph regularization. Smola & Kondor (2003) originally established that penalizing the discrete graph norm $\|f\|_L^2 = f^\top \mathbf{L} f$ induces a Reproducing Kernel Hilbert Space (RKHS) where the kernel is the pseudoinverse of the Laplacian. Our Theorem 2.1 formalizes this duality for the vector-valued case via a tensor product RKHS. We note that this structural result can essentially be inferred from the comprehensive review of vector-valued functions by Alvarez et al. Alvarez et al. (2012).

Following Smola & Kondor (2003), any positive semi-definite kernel on the vertices that is a function of the Laplacian can be written in the eigenbasis of $\mathbf{L}$ as $\mathbf{K}_G = \sum_{i=1}^n r(\lambda_i) \mathbf{q}_i \mathbf{q}_i^\top$ where $\{(\lambda_i, \mathbf{q}_i)\}_{i=1}^n$ are the eigenpairs of $\mathbf{L}$ and $r(\cdot) \geq 0$ is a spectral transfer function. Our choice $\mathbf{K}_G = (\mathbf{L} + \rho \mathbf{I})^{-1}$ corresponds to $r(\lambda) = 1/(\lambda + \rho)$, which is monotone decreasing and therefore shrinks high-frequency components more strongly, enforcing a smooth/homophilous prior. In principle, non-monotone or band-pass transfer functions $r$ can encode more complex, possibly non-smooth or heterophilous relations between users; analyzing such priors in the bandit setting is an interesting direction for future work.

More recent works in graph signal processing adopt related Laplacian-based constructions but do not use the induced RKHS norm as the main vehicle for analysis. Venkitaraman et al. (2020) obtain Gaussian Processes over graphs from a Laplacian prior, and Zhi et al. (2023) further generalize

this by learning a spectral filter $g(\boldsymbol{L})$ applied directly to the Laplacian. In both cases, the focus is on batch regression and signal reconstruction; the underlying regularizer can be characterized spectrally in terms of the transfer function associated with $g$, in the sense of Smola & Kondor (2003), but it is not the primary object of study. By contrast, in our work we commit to the specific Green's-function kernel $\boldsymbol{K}_G = (\boldsymbol{L} + \rho \boldsymbol{I})^{-1}$, which corresponds to the classical Dirichlet energy regularizer and enforces a homophilous prior. This choice yields a simple, explicit RKHS norm that we can track throughout the analysis and directly tie to the effective dimension and regret in the multi-user bandit setting.

**Graph-structure Bandits**   Graph-based bandit models are also relevant but conceptually distinct. In nonstochastic bandits with graph-structured feedback, a learner chooses an arm (node) and observes the losses of that arm and its neighbors in a feedback graph, interpolating between full-information and standard bandits Alon et al. (2017). Regret bounds in this line of work typically scale with graph-theoretic quantities such as the independence number $\alpha(G)$ or related observability parameters Alon et al. (2013). Follow-up studies on bandits with feedback graphs and graphical bandits refine these guarantees and extend them to stochastic settings, switching costs, adversarial corruptions, non-stationary environments, and contextual bandits, with regret controlled by parameters such as domination and weak-domination numbers, clique-cover and independence numbers, or maximum acyclic subgraph–type quantities Liu et al. (2018a;b); Arora et al. (2019); Lu et al. (2021); Zhang et al. (2023). In our setting, the user graph instead encodes prior correlation across user value functions through a Laplacian kernel; feedback remains strictly bandit (we only observe the reward of the chosen user–arm pair). Consequently, the graph enters our analysis only via the spectrum of the user kernel and the resulting effective dimension, rather than via such side-information parameters used in graphical bandit regret bounds.

**Collaborative Bandits**   Our approach is related to collaborative contextual bandits on graph, which exploit relations among users to accelerate learning. The collaborative contextual bandit Wu et al. (2016) uses a user adjacency graph to share context and reward information online, effectively adding a Laplacian-type regularizer to a linear contextual bandit model. Other works consider low-rank or factorization-based collaborative bandits, such as matrix-factorization bandits for interactive recommendation Wang et al. (2017) and collaborative filtering bandits that co-cluster users and items in a bandit framework Li et al. (2016). A complementary line of work studies multi-agent bandits over social networks, where multiple players observe or share each other's actions and rewards to reduce regretKolla et al. (2018); Chawla et al. (2023); Christakopoulou & Banerjee (2018). These methods typically either (i) impose linear models with manually chosen regularizers, or (ii) model collaboration via latent factors, clustering, or message passing, without an explicit multi-output RKHS / GP interpretation. By contrast, our Laplacian-kernelized construction provides a principled kernel view of collaboration: the known user graph defines a positive-definite user kernel that is combined with a flexible context kernel, leading to algorithms whose uncertainty quantification and regret depend explicitly on the joint spectrum of the graph Laplacian and the base kernel, rather than on the number of users, clusters, or latent dimensions.

**Cooperative Multi-Agent Kernelized Bandits**   Dubey et al. (2020) study a cooperative multi-agent kernelized contextual bandit with delayed communication over a fixed graph $G = (V, E)$. In their model, *every* agent $v \in V$ acts at every round $t$, selecting an action $x_{v,t}$ and receiving a reward $y_{v,t}$, so that after $T$ rounds there are $|V|T$ observations; the graph $G$ is used solely to constrain message passing and appears in the regret via graph-theoretic quantities (e.g., clique numbers of graph powers), but it does *not* enter the construction of the similarity kernel between agents or the modeling of the reward functions themselves. Instead, Dubey et al. posit a latent "network context" $z_v$ for each agent and assume a global function $F(x, z)$ in the RKHS of a product kernel $K((x, z), (x', z')) = K_x(x, x')K_z(z, z')$. When the network contexts (or the kernel $K_z$) are not available, they propose to *estimate* them from the contexts $x_{v,t}$ by embedding each agent's context distribution $P_v$ into the RKHS of $K_x$ and defining $K_z$ as an RBF kernel on these mean embeddings. Thus, the agent kernel is ultimately a learned similarity over (estimated) context distributions, and the underlying communication graph plays no direct role in defining task similarity or a smoothness penalty on $(f_v)_{v \in V}$.

By contrast, our setting follows the Gang-of-Bandits model: at each time step a single user is drawn at random, we choose one action for that user, and we observe only one reward, so that after $T$

rounds we have $T$ observations rather than $|V|T$. We also behave as a centralized learner rather than a decentralized network of bandits. Most importantly, we do not introduce or estimate any latent network contexts; instead, we assume a given user graph and *fix* the agent kernel to the inverse regularized Laplacian,

$$K_z(u, v) = [\boldsymbol{L}_\rho^{-1}]_{u,v}.$$

This kernel is tightly coupled to the global homophily penalty on the vector of reward functions and yields an explicit RKHS norm with a clear smoothness interpretation. This principled graph-based construction allows us to carry out a spectral analysis of the resulting multi-user kernel, relate the regret to the spectrum of $\boldsymbol{L}_\rho$, and highlight how the effective dimension adapts to the cluster structure of the user graph, rather than reducing network information to ad hoc latent features inferred from context distributions.

## B  PROOF OF THEOREM 2.1

*Proof.* The proof proceeds in three main steps: (1) We construct the Hilbert space for our multi-user problem as the tensor product of the user space and the context space; (2) We define a feature map into this space and show that its inner product yields the kernel $K$. This establishes that our constructed space is indeed the RKHS $\mathcal{H}$; (3) We characterize the elements of $\mathcal{H}$ and derive the expression for their norm.

**Step 1: Constructing the Hilbert Space via Tensor Product.** Let $\mathcal{H}_G = \mathbb{R}^n$ be the finite-dimensional Hilbert space for the users, equipped with the standard Euclidean inner product $\langle \boldsymbol{u}, \boldsymbol{v} \rangle_{\mathcal{H}_G} = \boldsymbol{u}^\top \boldsymbol{v}$. $\{\boldsymbol{e}_i\}_{i=1}^n$ forms the standard orthonormal basis for $\mathcal{H}_G$. Our *multi-user RKHS $\mathcal{H}$* is the tensor product of $\mathcal{H}_G$ and $\mathcal{H}_x$:

$$\mathcal{H} := \mathcal{H}_G \otimes \mathcal{H}_x = \mathbb{R}^n \otimes \mathcal{H}_x.$$

The elements of $\mathcal{H}$ are (limits of) finite linear combinations of elementary tensors of the form $\boldsymbol{u} \otimes h$, where $\boldsymbol{u} \in \mathcal{H}_U$ and $h \in \mathcal{H}_x$. The inner product in $\mathcal{H}$ is defined on these elementary tensors and extended by linearity:

$$\langle \boldsymbol{u}_1 \otimes h_1, \boldsymbol{u}_2 \otimes h_2 \rangle_{\mathcal{H}} := \langle \boldsymbol{u}_1, \boldsymbol{u}_2 \rangle_{\mathcal{H}_G} \langle h_1, h_2 \rangle_{\mathcal{H}_x}.$$

**Step 2: Defining the Feature Map and Verifying the Kernel.** Let $\boldsymbol{L}_\rho^{1/2}$ be the unique symmetric positive definite square root of $\boldsymbol{L}_\rho$. We define the feature map $\phi : (\mathcal{U} \times \mathcal{D}) \to \mathcal{H}$ as:

$$\phi(\boldsymbol{x}, u) := \left( L_\rho^{-1/2} \boldsymbol{e}_i \right) \otimes \varphi(\boldsymbol{x}).$$

This is a valid element of $\mathcal{H}$ since $\boldsymbol{L}_\rho^{-1/2} \boldsymbol{e}_i \in \mathbb{R}^n = \mathcal{H}_G$ and $\varphi(\boldsymbol{x}) \in \mathcal{H}_x$. Let's compute the inner product of two such feature mappings in $\mathcal{H}$:

$$\begin{aligned}
\langle \phi(\boldsymbol{x}, i), \phi(\boldsymbol{y}, j) \rangle_{\mathcal{H}} &= \langle (\boldsymbol{L}_\rho^{-1/2} \boldsymbol{e}_i) \otimes \varphi(\boldsymbol{x}), (\boldsymbol{L}_\rho^{-1/2} \boldsymbol{e}_j) \otimes \varphi(\boldsymbol{y}) \rangle_{\mathcal{H}} \\
&= \langle \boldsymbol{L}_\rho^{-1/2} \boldsymbol{e}_i, \boldsymbol{L}_\rho^{-1/2} \boldsymbol{e}_j \rangle_{\mathcal{H}_G} \cdot \langle \varphi(\boldsymbol{x}), \varphi(\boldsymbol{y}) \rangle_{\mathcal{H}_x} \\
&= \boldsymbol{e}_i^\top \boldsymbol{L}_\rho^{-1} \boldsymbol{e}_j \cdot K_x(\boldsymbol{x}, \boldsymbol{y}) \\
&= [\boldsymbol{L}_\rho^{-1}]_{ij} \cdot K_x(\boldsymbol{x}, \boldsymbol{y}) = K((\boldsymbol{x}, i), (\boldsymbol{y}, j)).
\end{aligned}$$

By the fundamental property of RKHS, since the kernel $K$ is generated by the inner product of the feature map $\phi$ in the Hilbert space $\mathcal{H}$, $\mathcal{H}$ is the unique RKHS associated with $K$.

**Step 3: Characterizing Functions in $\mathcal{H}$ and their Norms.** An element of $\mathcal{H}$ is a function $f : (\mathcal{U} \times \mathcal{D}) \to \mathbb{R}$. By the Riesz representation theorem, for each $f \in \mathcal{H}$, there exists a unique element $\boldsymbol{\theta} \in \mathcal{H}$ such that $f(\cdot, \cdot) = \langle \boldsymbol{\theta}, \phi(\cdot, \cdot) \rangle_{\mathcal{H}}$ and $\|f\|_{\mathcal{H}} = \|\boldsymbol{\theta}\|_{\mathcal{H}}$. For some component functions $\{g_k\}_{k=1}^n \subset \mathcal{H}_x$, we can uniquely express $\boldsymbol{\theta}$ as

$$\boldsymbol{\theta} = \sum_{k=1}^n \boldsymbol{e}_k \otimes g_k$$

and the squared norm of $\boldsymbol{\theta}$ in $\mathcal{H}$ is then:

$$\|\boldsymbol{\theta}\|_{\mathcal{H}}^2 = \Big\langle \sum_k \boldsymbol{e}_k \otimes g_k, \sum_l \boldsymbol{e}_l \otimes g_l \Big\rangle_{\mathcal{H}} = \sum_{k,l} \langle \boldsymbol{e}_k, \boldsymbol{e}_l \rangle_{\mathcal{H}_G} \langle g_k, g_l \rangle_{\mathcal{H}_x} = \sum_{k=1}^n \|g_k\|_{\mathcal{H}_x}^2.$$

Then we can relate our reward functions $f_{1:n}$ to the component functions $\{g_k\}_{k=1}^n$:

$$
\begin{aligned}
f_i(\boldsymbol{x}) &= \langle \boldsymbol{\theta}, \phi((\boldsymbol{x}, i)) \rangle_{\mathcal{H}} \\
&= \langle \sum_{k=1}^n \boldsymbol{e}_k \otimes g_k, (\boldsymbol{L}_\rho^{-1/2}\boldsymbol{e}_i) \otimes \varphi(\boldsymbol{x}) \rangle_{\mathcal{H}} \\
&= \sum_{k=1}^n \langle \boldsymbol{e}_k, \boldsymbol{L}_\rho^{-1/2}\boldsymbol{e}_i \rangle_{\mathcal{H}_G} \cdot \langle g_k, \varphi(\boldsymbol{x}) \rangle_{\mathcal{H}_x} \\
&= \sum_{k=1}^n [\boldsymbol{L}_\rho^{-1/2}]_{ki} \cdot \langle g_k, \varphi(\boldsymbol{x}) \rangle_{\mathcal{H}_x} \\
&= \sum_{k=1}^n [\boldsymbol{L}_\rho^{-1/2}]_{ki} g_k(\boldsymbol{x}) \quad (\text{since } \langle g, \varphi(\boldsymbol{x}) \rangle_{\mathcal{H}_x} = g(\boldsymbol{x}))
\end{aligned}
$$

which leads to

$$g_k(\boldsymbol{x}) = \sum_{j=1}^n [\boldsymbol{L}_\rho^{1/2}]_{kj} f_j(\boldsymbol{x}).$$

This equality holds for the functions as elements of $\mathcal{H}_\kappa$: $g_k = \sum_{j=1}^n [L_\rho^{1/2}]_{kj} f_j$.

Finally, we compute the norm of $f$ in $\mathcal{H}$:

$$
\begin{aligned}
\|f\|_{\mathcal{H}}^2 = \|\boldsymbol{\theta}\|_{\mathcal{H}}^2 &= \sum_{k=1}^n \|g_k\|_{\mathcal{H}_x}^2 \\
&= \sum_{k=1}^n \left\| \sum_{j=1}^n [\boldsymbol{L}_\rho^{1/2}]_{kj} f_j \right\|_{\mathcal{H}_x}^2 \\
&= \sum_{k=1}^n \langle \sum_{j=1}^n [\boldsymbol{L}_\rho^{1/2}]_{kj} f_j, \sum_{l=1}^n [\boldsymbol{L}_\rho^{1/2}]_{kl} f_l \rangle_{\mathcal{H}_x} \\
&= \sum_{k=1}^n \sum_{j,l=1}^n [L_\rho^{1/2}]_{kj} [\boldsymbol{L}_\rho^{1/2}]_{kl} \langle f_j, f_l \rangle_{\mathcal{H}_x} \\
&= \sum_{j,l=1}^n \left( \sum_{k=1}^n [\boldsymbol{L}_\rho^{1/2}]_{kj} [L_\rho^{1/2}]_{kl} \right) \langle f_j, f_l \rangle_{\mathcal{H}_x} \\
&= \sum_{j,l=1}^n [\boldsymbol{L}_\rho]_{jl} \langle f_j, f_l \rangle_{\mathcal{H}_x}
\end{aligned}
$$

where the last step is because the term in parentheses is the $(j, l)$-th element of the matrix product $(\boldsymbol{L}_\rho^{1/2})^\top \boldsymbol{L}_\rho^{1/2} = \boldsymbol{L}_\rho^{1/2} \boldsymbol{L}_\rho^{1/2} = \boldsymbol{L}_\rho$. By polarization identity, the associated inner product in $\mathcal{H}$ is:

$$\langle f, g \rangle_{\mathcal{H}} := \sum_{i,j=1}^n [\boldsymbol{L}_\rho]_{ij} \langle f_i, g_j \rangle_{\mathcal{H}_x}.$$

To see that $\|f\|_{\mathcal{H}}^2$ is the exactly the penalty in equation 1 , we expand $\boldsymbol{L}_\rho = \rho I_n + \boldsymbol{L}$:

$$
\begin{aligned}
\|f\|_{\mathcal{H}}^2 &= \sum_{j,l=1}^{n} (\rho\mathbb{I}\{j=l\} + [\boldsymbol{L}]_{jl})\langle f_j, f_l\rangle_{\mathcal{H}_x} \\
&= \rho\sum_{j=1}^{n} \|f_j\|_{\mathcal{H}_x}^2 + \sum_{j,l=1}^{n} [\boldsymbol{L}]_{jl}\langle f_j, f_l\rangle_{\mathcal{H}_x}.
\end{aligned}
$$

Using the standard identity for the Laplacian quadratic form, the second term in the above equation is exactly $\frac{1}{2}\sum_{i,j} w_{ij}\|f_i - f_j\|_{\mathcal{H}_x}^2$, we get:

$$
\|f\|_{\mathcal{H}}^2 = \rho\sum_{j=1}^{n} \|f_j\|_{\mathcal{H}_x}^2 + \frac{1}{2}\sum_{i,j} w_{ij}\|f_i - f_j\|_{\mathcal{H}_x}^2.
$$

This completes the proof. $\qquad\square$

## C  PROOFS IN ANALYSIS

We first define following additional notations

$$
\begin{aligned}
\boldsymbol{\Phi}_t &:= [\phi(\boldsymbol{x}_1, u_1), \cdots, \phi(\boldsymbol{x}_t, u_t)]^\top & (15) \\
\boldsymbol{J}_t &:= \boldsymbol{\Phi}_t^\top \boldsymbol{\Phi}_t & (16) \\
\boldsymbol{\Gamma}_t &:= \boldsymbol{J}_t + \lambda\boldsymbol{I}_\infty & (17) \\
\boldsymbol{\Sigma}_t &:= \boldsymbol{K}_t + \lambda\boldsymbol{I}_t & (18)
\end{aligned}
$$

Here we have $\boldsymbol{\Phi}_t \in \mathbb{R}^{t\times\infty}$ and $\boldsymbol{J}_t, \boldsymbol{\Gamma}_t$ are from $\mathbb{R}^{\infty\times\infty}$.

Then we define some useful events for concentration:

$$
\begin{aligned}
\mathcal{E}_t^{\text{ts}} &= \{|z_t(\boldsymbol{x})| \le \sqrt{2\log(t^2|\mathcal{D}_t|)}, \text{ for all } \boldsymbol{x} \in \mathcal{D}_t\} \\
\mathcal{E}_t^a &= \{\mu_{u_t,t-1}(\boldsymbol{x}_t^*) + \beta_t z_t(\boldsymbol{x}_t^*)\sigma_{u_t,t-1}(\boldsymbol{x}_t^*) > f(\boldsymbol{x}_t^*, u_t)\}
\end{aligned}
$$

where $z_t(\boldsymbol{x}) \sim \mathcal{N}(0,1)$ stands for the resampling randomness in Thompson Sampling. We also define the confidence set at round $t$:

$$
\mathcal{C}_t := \{|\mu_{u_t,t-1}(\boldsymbol{x}_t) - f(\boldsymbol{x}_t, u_t)| \le \beta_t \cdot \sigma_{u_t,t-1}(\boldsymbol{x}_t)\} \tag{19}
$$

where

$$
\beta_t := \left( B_\rho + \sqrt{\frac{\sigma^2}{\lambda} \cdot \log\det(\boldsymbol{I}_{t-1} + \lambda^{-1}\boldsymbol{K}_{t-1}) + \frac{2\sigma^2}{\lambda}\log\frac{1}{\delta}} \right).
$$

In addition, recall the following effective dimension

$$
\tilde{d} := \frac{\log\det(\boldsymbol{I}_T + \boldsymbol{K}_T/\lambda)}{\log(1 + TK_{\max}/\lambda)}
$$

and the upper bound of the optimality gap:

$$
|\Delta_t| \le B_\Delta := 2B_\rho K_{\max}^{1/2}.
$$

Lastly, we provide the following Lemmas, which are commonly required in regret analysis.

**Lemma C.1** (Concentrations for TS). *For all $t \in [T]$, we have $\mathbb{P}_t(\bar{\mathcal{E}}_t^{\text{ts}}) \le t^{-2}$ and $\mathbb{P}_t(\mathcal{E}_t^a|\mathcal{C}_t) \ge (4e\sqrt{\pi})^{-1}$.*

**Lemma C.2** (One Step Regret Bound for TS). *Suppose $\mathbb{P}_t(\mathcal{E}_t^a) - \mathbb{P}_t(\bar{\mathcal{E}}_t^{\text{ts}}) > 0$. Then for any $t$, almost surely,*

$$
\mathbb{E}_t[\Delta_t\mathbb{I}_{\mathcal{C}_t}] \le \mathbb{I}_{\mathcal{C}_t} \cdot \left\{ \left( \frac{2}{\mathbb{P}_t(\mathcal{E}_t^a) - \mathbb{P}_t(\bar{\mathcal{E}}_t^{\text{ts}})} + 1 \right) \cdot \mathbb{E}_t[\gamma_t\sigma_{u_t,t-1}(\boldsymbol{x}_t)] + B_\Delta \cdot \mathbb{P}_t(\bar{\mathcal{E}}_t^{\text{ts}}) \right\}
$$

*where $\gamma_t := \beta_t + \beta_t\sqrt{2\log(t^2|\mathcal{D}_t|)}$ and $B_\Delta := 2B_\rho K_{\max}^{1/2}$*

**Lemma C.3** (Cumulative Uncertainty Bound). *We have the upper bound for the cumulative estimated uncertainty:*

$$\sum_{t=1}^{T} \sigma_{u_t,t-1}(\boldsymbol{x}_t) \leq \sqrt{2T \max\{1, K_{\max}\} \cdot \log \det(\boldsymbol{I}_T + \lambda^{-1}\boldsymbol{K}_T)}$$

**Lemma C.4** (Dual Identities). *With the defined notations in equation 15, we have two key identities:*

$$\boldsymbol{\Sigma}_t^{-1}\boldsymbol{\Phi}_t = \boldsymbol{\Phi}_t\boldsymbol{\Gamma}_t^{-1}, \text{ and } \sigma_{u,t}^2(\boldsymbol{x}) = \lambda\|\phi(\boldsymbol{x},u)\|_{\boldsymbol{\Gamma}_t^{-1}}^2.$$

## C.1 Proof of Confidence Set

*Proof of Theorem 4.1.* We first decompose

$$\begin{aligned}
\mu_{u,t}(\boldsymbol{x}) - f(\boldsymbol{x},u) &= \boldsymbol{k}_t(\boldsymbol{x},u)^\top \boldsymbol{\Sigma}_t^{-1}(\boldsymbol{\Phi}_t\boldsymbol{\theta} + \boldsymbol{\epsilon}_t) - \boldsymbol{\theta}^\top\phi(\boldsymbol{x},u) \\
&= (\boldsymbol{\Phi}_t^\top\boldsymbol{\Sigma}_t^{-1}\boldsymbol{k}_t(\boldsymbol{x},u))^\top\boldsymbol{\theta} + \boldsymbol{k}_t(\boldsymbol{x},u)^\top\boldsymbol{\Sigma}_t^{-1}\boldsymbol{\epsilon}_t - \boldsymbol{\theta}^\top\phi(\boldsymbol{x},u) \\
&= \underbrace{\langle\boldsymbol{\theta}, \delta_t(\boldsymbol{x},u)\rangle}_{\text{bias}_t(\boldsymbol{x},u)} + \underbrace{\boldsymbol{k}_t(\boldsymbol{x},u)^\top\boldsymbol{\Sigma}_t^{-1}\boldsymbol{\epsilon}_t}_{\text{noise}_t(\boldsymbol{x},u)}
\end{aligned}$$

where $\delta_t(\boldsymbol{x},u) = \boldsymbol{\Phi}_t^\top\boldsymbol{\Sigma}_t^{-1}\boldsymbol{k}_t(\boldsymbol{x},u) - \phi(\boldsymbol{x},u) \in \ell^2$. Our target is to bound the $\text{bias}_t(\boldsymbol{x},u)$ and $\text{noise}_t(\boldsymbol{x},u)$. We state the following Lemmas:

**Lemma C.5** (Bias Identity). *The squared bias is the degraded variance for noise:*

$$\|\delta_t(\boldsymbol{x},u)\|_{\ell^2}^2 = \sigma_{u,t}^2(\boldsymbol{x}) - \lambda\boldsymbol{k}_t(\boldsymbol{x},u)^\top\boldsymbol{\Sigma}_t^{-2}\boldsymbol{k}_t(\boldsymbol{x},u) \tag{20}$$

*In particular, we have* $\|\delta_t(\boldsymbol{x},u)\|_{\ell^2} \leq \sigma_{u,t}(\boldsymbol{x})$ *and* $\lambda\boldsymbol{k}_t(\boldsymbol{x},u)^\top\boldsymbol{\Sigma}_t^{-2}\boldsymbol{k}_t(\boldsymbol{x},u) < \sigma_{u,t}^2(\boldsymbol{x})$.

**Lemma C.6** (Noise Bound). *With high probability, we have the upper bound for the following norm of noise vector* $\boldsymbol{\epsilon}_t$:

$$\|\boldsymbol{\Phi}_t\boldsymbol{\epsilon}_t\|_{\boldsymbol{\Gamma}_t^{-1}} \leq \sqrt{\sigma^2 \log\det(\boldsymbol{I}_t + \lambda^{-1}\boldsymbol{K}_t) + 2\sigma^2\log\frac{1}{\delta}}$$

From Lemma C.5, we could bound the bias by

$$\text{bias}_t(\boldsymbol{x},u) \leq \|\boldsymbol{\theta}\|_{\ell^2}\|\delta_t(\boldsymbol{x},u)\|_{\ell^2} \leq B_\rho\sigma_{u,t}(\boldsymbol{x}). \tag{21}$$

Using the identities in above Lemma C.4, we note that

$$\begin{aligned}
\text{noise}_t(\boldsymbol{x},u) &= \boldsymbol{k}_t(\boldsymbol{x},u)^\top\boldsymbol{\Sigma}_t^{-1}\boldsymbol{\epsilon}_t \\
&= \phi(\boldsymbol{x},u)^\top\boldsymbol{\Gamma}_t^{-1}\boldsymbol{\Phi}_t\boldsymbol{\epsilon}_t \\
&= \langle\phi(\boldsymbol{x},u), \boldsymbol{\Phi}_t\boldsymbol{\epsilon}_t\rangle_{\boldsymbol{\Gamma}_t^{-1}} \\
&\leq \|\phi(\boldsymbol{x},u)\|_{\boldsymbol{\Gamma}_t^{-1}} \cdot \|\boldsymbol{\Phi}_t\boldsymbol{\epsilon}_t\|_{\boldsymbol{\Gamma}_t^{-1}} \\
&= \frac{\sigma_{u,t}(\boldsymbol{x})}{\sqrt{\lambda}} \cdot \|\boldsymbol{\Phi}_t\boldsymbol{\epsilon}_t\|_{\boldsymbol{\Gamma}_t^{-1}}
\end{aligned}$$

where the inequality is from the Cauchy-Schwarz inequality for the inner product $\langle\cdot,\cdot\rangle_{\boldsymbol{\Gamma}_t^{-1}}$.

Our Lemma C.6 gives the high probability upper bound for the norm $\|\boldsymbol{\Phi}_t\boldsymbol{\epsilon}_t\|_{\boldsymbol{\Gamma}_t^{-1}}$, leading to

$$\text{noise}_t(\boldsymbol{x},u) \leq \frac{\sigma_{u,t}(\boldsymbol{x})}{\sqrt{\lambda}} \cdot \sqrt{\sigma^2\log\det(\boldsymbol{I}_t + \lambda^{-1}\boldsymbol{K}_t) + 2\sigma^2\log\frac{1}{\delta}} \tag{22}$$

Now combine equation 21 and equation 22 together, we have

$$\begin{aligned}
|\mu_{u,t}(\boldsymbol{x}) - f(\boldsymbol{x},u)| &\leq |\text{bias}_t(\boldsymbol{x},u)| + |\text{noise}_t(\boldsymbol{x},u)| \\
&\leq \sigma_{u,t}(\boldsymbol{x})\left(B_\rho + \sqrt{\frac{\sigma^2}{\lambda} \cdot \log\det(\boldsymbol{I}_t + \lambda^{-1}\boldsymbol{K}_t) + \frac{2\sigma^2}{\lambda}\log\frac{1}{\delta}}\right).
\end{aligned}$$

$\square$

## C.2 PROOF OF REGRET BOUND OF `LK-GP-UCB`

*Proof of Theorem 4.2.* Recall the instantaneous regret at time $t$ is $\Delta_t = f(\boldsymbol{x}_t^*, u_t) - f(\boldsymbol{x}_t, u_t)$ and the cumulative regret in a time horizon $T$ is $\mathcal{R}_T = \sum_{t=1}^T \Delta_t$. We note event $\mathcal{C}_t := \{|\mu_{u_t, t-1}(\boldsymbol{x}_t) - f(\boldsymbol{x}_t, u_t)| \le \beta_t \cdot \sigma_{u_t, t-1}(\boldsymbol{x}_t)\}$ happens with high probability $(1 - \delta)$, according to Theorem 4.1,

$$\beta_t := \left( B_\rho + \sqrt{\frac{\sigma^2}{\lambda} \cdot \log\det(\boldsymbol{I}_{t-1} + \lambda^{-1}\boldsymbol{K}_{t-1}) + \frac{2\sigma^2}{\lambda}\log\frac{1}{\delta}} \right) \tag{23}$$

By Theorem 4.1, for all $t \ge 2$ with probability at least $1 - \delta$,

$$\begin{aligned}
\Delta_t = f(\boldsymbol{x}_t^*, u_t) - f(\boldsymbol{x}_t, u_t) &\le \mu_{u_t, t-1}(\boldsymbol{x}_t^*) + \beta_t \sigma_{u_t, t-1}(\boldsymbol{x}_t^*) - f(\boldsymbol{x}_t, u_t) \\
&\le \mu_{u_t, t-1}(\boldsymbol{x}_t) + \beta_t \sigma_{u_t, t-1}(\boldsymbol{x}_t) - f(\boldsymbol{x}_t, u_t) \\
&\le 2\beta_t \sigma_{u_t, t-1}(\boldsymbol{x}_t).
\end{aligned}$$

Thus we have high probability bound for the cumulative regret

$$\mathcal{R}_T \le 2\mathbb{E}\left[\beta_t \sum_{t=2}^T \sigma_{u_t, t-1}(\boldsymbol{x}_t)\right] + B_\Delta.$$

Then we apply Lemma C.3 and the definition of effective dimension in equation 11

$$\begin{aligned}
\sum_{t=1}^T \sigma_{u_t, t-1}(\boldsymbol{x}_t) &\le \sqrt{2T \max\{1, K_{\max}\} \cdot \log\det(\boldsymbol{I}_T + \lambda^{-1}\boldsymbol{K}_T)} \\
&= \sqrt{2T \max\{1, K_{\max}\} \cdot \tilde{d}\log(1 + T\lambda^{-1}K_{\max})}.
\end{aligned}$$

Therefore, we have the final high probability upper bound for regret:

$$\mathcal{R}_T \le 2\mathbb{E}[\beta_T]\sqrt{2T \max\{1, K_{\max}\} \cdot \tilde{d}\log(1 + T\lambda^{-1}K_{\max})} + B_\Delta.$$

The next step is to analyze the order of the upper bound. By using the effective dimension $\tilde{d}$ again and dropping constants, we have

$$\begin{aligned}
\beta_t &\le B_\rho + \sqrt{\frac{\sigma^2}{\lambda} \cdot \tilde{d}\log(1 + T\lambda^{-1}K_{\max}) + \frac{2\sigma^2}{\lambda}\log\frac{1}{\delta}} = \mathcal{O}(\sqrt{\tilde{d}\log(T)}) \\
\Rightarrow \mathcal{R}_T &= \mathcal{O}(\tilde{d}\log(T)\sqrt{T}) = \tilde{\mathcal{O}}(\tilde{d}\sqrt{T}).
\end{aligned}$$

$\square$

## C.3 PROOF OF REGRET BOUND OF `LK-GP-TS`

*Proof of Theorem 4.3.* We start from the decomposition of the cumulative regret

$$\mathcal{R}_T = \sum_{t=1}^T \mathbb{E}[\Delta_t] = \sum_{t=1}^T \mathbb{E}[\Delta_t \mathbb{I}_{\mathcal{C}_t}] + \sum_{t=1}^T \mathbb{E}[\Delta_t \mathbb{I}_{\bar{\mathcal{C}}_t}].$$

By Theorem 4.1 and the upper bound for the optimality gap, we know the second term is bounded:

$$\sum_{t=1}^T \mathbb{E}[\Delta_t \mathbb{I}_{\bar{\mathcal{C}}_t}] \le \delta B_\Delta$$

by letting $\mathbb{P}(\mathcal{C}_t) \le \delta/T$ for all $t$ in Theorem 4.1.

For the regret on the event $\mathcal{C}_t$, by Lemma C.2, almost surely, we have

$$\mathbb{E}_t[\Delta_t \mathbb{I}_{\mathcal{C}_t}] \le \mathbb{I}_{\mathcal{C}_t} \cdot \left\{ \left( \frac{2}{\mathbb{P}_t(\mathcal{E}_t^a) - \mathbb{P}_t(\bar{\mathcal{E}}_t^{\text{ts}})} + 1 \right) \cdot \mathbb{E}_t[\gamma_t \sigma_{u_t, t-1}(\boldsymbol{x}_t)] + B_\Delta \cdot \mathbb{P}_t(\bar{\mathcal{E}}_t^{\text{ts}}) \right\}$$

where $\gamma_t := \beta_t + \beta_t\sqrt{2\log(t^2|\mathcal{D}_t|)}$. Note that $\mathbb{P}_t(\mathcal{E}_t^a) - \mathbb{P}_t(\bar{\mathcal{E}}_t^{\mathrm{ts}}) \geq \frac{1}{4e\sqrt{\pi}} - \frac{1}{t^2} \geq \frac{1}{20e\sqrt{\pi}}$ by Lemma C.1 and the fact that $t^2 \geq 5e\sqrt{\pi}$ for all $t \geq 5$. Thus we have

$$\mathbb{E}_t[\Delta_t\mathbb{I}_{\mathcal{C}_t}] \leq \mathbb{I}_{\mathcal{C}_t} \cdot \left\{ 194\mathbb{E}_t[\gamma_t\sigma_{u_t,t-1}(\boldsymbol{x}_t)] + B_\Delta t^{-2} \right\}$$

by using $40e\sqrt{\pi} + 1 \leq 194$. Taking summation on both side for our target cumulative regret, we get

$$\sum_{t=1}^T \mathbb{E}[\Delta_t\mathbb{I}_{\mathcal{C}_t}] = \mathbb{E}[\sum_{t=1}^T \mathbb{E}_t[\Delta_t\mathbb{I}_{\mathcal{C}_t}]]$$

$$\leq \mathbb{E}[\sum_{t=5}^T \left( 194\mathbb{E}_t[\gamma_t\sigma_{u_t,t-1}(\boldsymbol{x}_t)] + B_\Delta t^{-2} \right) + 4B_\Delta]$$

$$\leq \mathbb{E}[194\sum_{t=5}^T \mathbb{E}_t[\gamma_t\sigma_{u_t,t-1}(\boldsymbol{x}_t)] + (4 + \frac{\pi^2}{6})B_\Delta]$$

$$\leq \mathbb{E}[194\gamma_T\mathbb{E}_t[\sum_{t=1}^T \sigma_{u_t,t-1}(\boldsymbol{x}_t)] + (4 + \frac{\pi^2}{6})B_\Delta]$$

where the second equality is using $\sum_{t=1}^\infty t^{-2} = \pi^2/6$ and the last step is from the monotonicity of the $\gamma_t$ and the nonnegative of $\sigma_{u,t}(\boldsymbol{x})$. Our next focus is bounding the summation of uncertainty. As the same approach in the proof of Theorem 4.2, we apply Lemma C.3 and the definition of effective dimension in equation 11

$$\sum_{t=1}^T \sigma_{u_t,t-1}(\boldsymbol{x}_t) \leq \sqrt{2T\max\{1, K_{\max}\} \cdot \log\det(\boldsymbol{I}_T + \lambda^{-1}\boldsymbol{K}_T)}$$

$$= \sqrt{2T\max\{1, K_{\max}\} \cdot \tilde{d}\log(1 + T\lambda^{-1}K_{\max})}.$$

Thus we have

$$\sum_{t=1}^T \mathbb{E}[\Delta_t\mathbb{I}_{\mathcal{C}_t}] \leq 194\mathbb{E}[\gamma_T]\sqrt{2T\max\{1, K_{\max}\} \cdot \tilde{d}\log(1 + T\lambda^{-1}K_{\max})} + (4 + \frac{\pi^2}{6})B_\Delta$$

leading to the high probability $(1 - \delta)$ regret upper bound:

$$\mathcal{R}_T \leq 194\mathbb{E}[\gamma_T]\sqrt{2T\max\{1, K_{\max}\} \cdot \tilde{d}\log(1 + T\lambda^{-1}K_{\max})} + (4 + \frac{\pi^2}{6})B_\Delta + \delta B_\Delta.$$

For the order of the upper bound, we first analyze $\mathbb{E}[\gamma_T]$, by using the definition of effective dimension $\tilde{d}$ again and dropping constants

$$\gamma_T \leq \left( 1 + \sqrt{2\log(T^2 M)} \right) \cdot \left( B_\rho + \sqrt{\frac{\sigma^2}{\lambda} \cdot \tilde{d}\log(1 + T\lambda^{-1}K_{\max}) + \frac{2\sigma^2}{\lambda}\log\frac{1}{\delta}} \right)$$

$$= \mathcal{O}(\log(T)\sqrt{\tilde{d}}).$$

where $M$ is the upper bound for the size of action set at time $t$, i.e. $|\mathcal{D}_t| \leq M$ for all $t \leq T$. Therefore,

$$\mathcal{R}_T = \mathcal{O}(\tilde{d}\log(T)^{3/2}\sqrt{T}) = \tilde{\mathcal{O}}(\tilde{d}\sqrt{T}).$$

$\square$

# D PROOF OF LEMMAS

## D.1 PROOF OF LEMMA C.1

*Proof.* Using the standard Gaussian tail bound and the classical union bound, we have

$$\mathbb{P}_t(|z_t(\boldsymbol{x})| > u) \leq |\mathcal{D}_t|e^{-u^2/2}.$$

By letting $u = \sqrt{2 \log(t^2 |\mathcal{D}_t|)}$, we obtain $\mathbb{P}_t(\bar{\mathcal{E}}_t^{ts}) \le t^{-2}$.

For the result of event $\mathcal{E}_t^a$, we have

$$
\begin{aligned}
\mathbb{P}_t\left(\mu_{u_t,t-1}(\boldsymbol{x}_t^*) + \beta_t z_t(\boldsymbol{x}_t^*)\sigma_{u_t,t-1}(\boldsymbol{x}_t^*) > f(\boldsymbol{x}_t^*, u_t)|\mathcal{C}_t\right) &= \mathbb{P}_t\left(z_t(\boldsymbol{x}_t^*) > \frac{f(\boldsymbol{x}_t^*, u_t) - \mu_{u_t,t-1}(\boldsymbol{x}_t^*)}{\beta_t \sigma_{u_t,t-1}(\boldsymbol{x}_t^*)}|\mathcal{C}_t\right) \\
&\ge \mathbb{P}_t(z_t(\boldsymbol{x}_t^*) > 1) \\
&\ge (4e\sqrt{\pi})^{-1}
\end{aligned}
$$

where the first inequality is from the fact that $\mathcal{C}_t$ holds and the last step is directly obtain by the fact that $\mathbb{P}(Z \ge 1) \ge (4e\sqrt{\pi})^{-1}$ for $Z \sim \mathcal{N}(0,1)$.

$\square$

### D.2 Proof of Lemma C.2

*Proof.* This proof is following the classical analysis for Thompson Sampling algorithms Kveton et al. (2019); Wu et al. (2022); Wu & Amini (2024).

We first recall $\mathbb{E}_t[\cdot] = \mathbb{E}[\cdot|\mathcal{F}_t]$. Given the randomness from the history $\mathcal{F}_t$, event $\mathcal{C}_t$ becomes deterministic and the randomness is only from the resampling step. So we have

$$
\begin{aligned}
\mathbb{E}_t[\Delta_t \mathbb{I}_{\mathcal{C}_t}] &= \mathbb{I}_{\mathcal{C}_t} \cdot \mathbb{E}_t[\Delta_t] \\
&= \mathbb{I}_{\mathcal{C}_t} \cdot \left(\mathbb{E}_t[\Delta_t \mathbb{I}_{\mathcal{E}_t^{ts}}] + \mathbb{E}_t[\Delta_t \mathbb{I}_{\bar{\mathcal{E}}_t^{ts}}]\right) \\
&\le \mathbb{I}_{\mathcal{C}_t} \cdot \left(\mathbb{E}_t[\Delta_t \mathbb{I}_{\mathcal{E}_t^{ts}}] + B_\Delta \cdot \mathbb{P}_t(\bar{\mathcal{E}}_t^{ts})\right)
\end{aligned}
$$

where the last step is from the boundness of the optimality gap $\Delta_t \le B_\Delta$. Our following focus is bounding $\mathbb{E}_t[\Delta_t \mathbb{I}_{\mathcal{E}_t^{ts}}]$, indicating $\mathcal{C}_t$ holds in the remaining part of proof.

We then define the concept of "least uncertain undersampled" action, which is called unsaturated actions, defined as

$$
\mathcal{U}_t := \{\boldsymbol{x} \in \mathcal{D}_t : f(\boldsymbol{x}_t^*, u_t) < f(\boldsymbol{x}, u_t) + \gamma_t \sigma_{u_t,t-1}(\boldsymbol{x})\}
$$

where

$$
\gamma_t := \beta_t + \beta_t \sqrt{2 \log(t^2 |\mathcal{D}_t|)}
$$

and let $\bar{\boldsymbol{x}}_t$ be the least uncertain unsaturated action at time $t$:

$$
\bar{\boldsymbol{x}}_t = \arg\min_{\boldsymbol{x} \in \mathcal{U}_t} \gamma_t \sigma_{u_t,t-1}(\boldsymbol{x}).
$$

Recall the notation for the resampled index is $\tilde{\mu}_t(\boldsymbol{x}) = \mu_{u_t,t-1}(\boldsymbol{x}) + \beta_t z_t(\boldsymbol{x})\sigma_{u_t,t-1}(\boldsymbol{x})$. On the good situation $\mathcal{C}_t \cap \mathcal{E}_t^{ts}$, we have

$$
|\tilde{\mu}_t(\boldsymbol{x}) - f(\boldsymbol{x}, u_t)| \le |\tilde{\mu}_t(\boldsymbol{x}) - \mu_{u_t,t-1}(\boldsymbol{x})| + |\mu_{u_t,t-1}(\boldsymbol{x}) - f(\boldsymbol{x}, u_t)| \le \gamma_t \sigma_{u_t,t-1}(\boldsymbol{x}).
$$

Thus we can provide an initial upper bound for regret

$$
\begin{aligned}
\Delta_t &= f(\boldsymbol{x}_t^*, u_t) - f(\boldsymbol{x}_t, u_t) \\
&= f(\boldsymbol{x}_t^*, u_t) - f(\bar{\boldsymbol{x}}_t, u_t) + f(\bar{\boldsymbol{x}}_t, u_t) - f(\boldsymbol{x}_t, u_t) \\
&\le \gamma_t \sigma_{u_t,t-1}(\bar{\boldsymbol{x}}_t) + f(\bar{\boldsymbol{x}}_t, u_t) - f(\boldsymbol{x}_t, u_t) + \tilde{\mu}_t(\boldsymbol{x}_t) - \tilde{\mu}_t(\boldsymbol{x}_t) \quad (\text{by } \bar{\boldsymbol{x}}_t \in \mathcal{U}_t) \\
&\le 2\gamma_t \sigma_{u_t,t-1}(\bar{\boldsymbol{x}}_t) + \gamma_t \sigma_{u_t,t-1}(\boldsymbol{x}_t) + \tilde{\mu}_t(\bar{\boldsymbol{x}}_t) - \tilde{\mu}_t(\boldsymbol{x}_t) \quad (\text{ since } \mathcal{C}_t \cap \mathcal{E}_t^{ts}) \\
&\le 2\gamma_t \sigma_{u_t,t-1}(\bar{\boldsymbol{x}}_t) + \gamma_t \sigma_{u_t,t-1}(\boldsymbol{x}_t) \quad (\text{ by } \tilde{\mu}_t(\bar{\boldsymbol{x}}_t) < \tilde{\mu}_t(\boldsymbol{x}_t)).
\end{aligned}
\quad (24)
$$

Note that

$$
\gamma_t \sigma_{u_t,t-1}(\bar{\boldsymbol{x}}_t) \mathbb{I}\{\boldsymbol{x}_t \in \mathcal{U}_t\} \le \gamma_t \sigma_{u_t,t-1}(\boldsymbol{x}_t)
$$

and by taking $\mathbb{E}_t[\cdot]$ after multiplying both sides by $\mathbb{I}_{\mathcal{E}_t^{ts}}$, we have

$$
\sigma_{u_t,t-1}(\bar{\boldsymbol{x}}_t) \mathbb{P}_t(\{\boldsymbol{x}_t \in \mathcal{U}_t\} \cap \mathcal{E}_t^{ts}) \le \mathbb{E}_t[\sigma_{u_t,t-1}(\boldsymbol{x}_t)\mathbb{I}_{\mathcal{E}_t^{ts}}].
$$

Thus it remains to bound the probability $\mathbb{P}_t(\{\boldsymbol{x}_t \in \mathcal{U}_t\} \cap \mathcal{E}_t^{ts})$ from below.

We notice the following two facts. First, if $\tilde{\mu}_t(\boldsymbol{x}_t^*) > \tilde{\mu}_t(\boldsymbol{x})$ for all $\boldsymbol{x} \in \bar{\mathcal{U}}_t$, then $\boldsymbol{x}_t$ must belong to $\mathcal{U}_t$, which means $\{\tilde{\mu}_t(\boldsymbol{x}_t^*) > \max_{\boldsymbol{x} \in \bar{\mathcal{U}}_t} \tilde{\mu}_t(\boldsymbol{x})\} \subseteq \{\boldsymbol{x}_t \in \mathcal{U}_t\}$. Second, for any $\boldsymbol{x} \in \bar{\mathcal{U}}_t$, on the good situation $\mathcal{C}_t \cap \mathcal{E}_t^{\text{ts}} \cap \mathcal{E}_t^a$, we have

$$\tilde{\mu}_t(\boldsymbol{x}) \leq f(\boldsymbol{x}, u_t) + \gamma_t \sigma_{u_t, t-1}(\boldsymbol{x}) \leq f(\boldsymbol{x}_t^*, u_t) < \tilde{\mu}_t(\boldsymbol{x}^*)$$

which leads to $\mathcal{E}_t^a \subseteq \{\tilde{\mu}_t(\boldsymbol{x}_t^*) > \max_{\boldsymbol{x} \in \bar{\mathcal{U}}_t} \tilde{\mu}_t(\boldsymbol{x})\}$

Therefore, on event $\mathcal{C}_t$, we have

$$\begin{aligned}
\mathbb{P}_t(\{\boldsymbol{x}_t \in \mathcal{U}_t\} \cap \mathcal{E}_t^{\text{ts}}) &\geq \mathbb{P}_t(\{\tilde{\mu}_t(\boldsymbol{x}_t^*) > \max_{\boldsymbol{x} \in \bar{\mathcal{U}}_t} \tilde{\mu}_t(\boldsymbol{x})\} \cap \mathcal{E}_t^{\text{ts}}) \\
&\geq \mathbb{P}_t(\mathcal{E}_t^a \cap \mathcal{E}_t^{\text{ts}}) \\
&\geq \mathbb{P}_t(\mathcal{E}_t^a) - \mathbb{P}_t(\bar{\mathcal{E}}_t^{\text{ts}})
\end{aligned}$$

Now we have a upper bound for $\sigma_{u_t, t-1}(\bar{\boldsymbol{x}}_t)$:

$$\sigma_{u_t, t-1}(\bar{\boldsymbol{x}}_t) \leq \frac{\mathbb{E}_t[\sigma_{u_t, t-1}(\boldsymbol{x}_t) \mathbb{I}_{\mathcal{E}_t^{\text{ts}}}]}{\mathbb{P}_t(\{\boldsymbol{x}_t \in \mathcal{U}_t\} \cap \mathcal{E}_t^{\text{ts}})} \leq \frac{\mathbb{E}_t[\sigma_{u_t, t-1}(\boldsymbol{x}_t)]}{\mathbb{P}_t(\mathcal{E}_t^a) - \mathbb{P}_t(\bar{\mathcal{E}}_t^{\text{ts}})}$$

which gives the upper bound for instantaneous regret by plugging above result in equation 24:

$$\mathbb{E}_t[\Delta_t \mathbb{I}_{\mathcal{E}_t^{\text{ts}}}] \leq \left(\frac{2}{\mathbb{P}_t(\mathcal{E}_t^a) - \mathbb{P}_t(\bar{\mathcal{E}}_t^{\text{ts}})} + 1\right) \cdot \mathbb{E}_t[\gamma_t \sigma_{u_t, t-1}(\boldsymbol{x}_t)].$$

Therefore,

$$\mathbb{E}_t[\Delta_t \mathbb{I}_{\mathcal{C}_t}] \leq \mathbb{I}_{\mathcal{C}_t} \cdot \left\{\left(\frac{2}{\mathbb{P}_t(\mathcal{E}_t^a) - \mathbb{P}_t(\bar{\mathcal{E}}_t^{\text{ts}})} + 1\right) \cdot \mathbb{E}_t[\gamma_t \sigma_{u_t, t-1}(\boldsymbol{x}_t)] + B_\Delta \cdot \mathbb{P}_t(\bar{\mathcal{E}}_t^{\text{ts}})\right\}$$

□

### D.3 PROOF OF LEMMA C.3

*Proof.* We first apply Cauchy-Schwartz inequality and obtain

$$\sum_{t=1}^T \sigma_{u_t, t-1}(\boldsymbol{x}_t) \leq \sqrt{T \sum_{t=1}^T \sigma_{u_t, t-1}^2(\boldsymbol{x}_t)} = \sqrt{\lambda T \sum_{t=1}^T \frac{\sigma_{u_t, t-1}^2(\boldsymbol{x}_t)}{\lambda}}.$$

If $\lambda \geq K_{\max}$, using $\sigma_{u_t, t-1}^2(\boldsymbol{x}_t) \leq |K((\boldsymbol{x}_t, u_t)(\boldsymbol{x}_t, u_t))| \leq K_{\max}$, we know $\frac{\sigma_{u_t, t-1}^2(\boldsymbol{x}_t)}{\lambda} \leq 1$, which leads to

$$\sum_{t=1}^T \frac{\sigma_{u_t, t-1}^2(\boldsymbol{x}_t)}{\lambda} \leq 2 \sum_{t=1}^T \log\left(1 + \frac{1}{\lambda} \sigma_{u_t, t-1}^2(\boldsymbol{x}_t)\right) \leq \frac{2K_{\max}}{\lambda} \sum_{t=1}^T \log\left(1 + \frac{1}{\lambda} \sigma_{u_t, t-1}^2(\boldsymbol{x}_t)\right)$$

by applying the fact that $x \leq 2\log(1+x)$ if $x \leq 1$.

If $\lambda \leq K_{\max}$, still using $\sigma_{u_t, t-1}^2(\boldsymbol{x}_t) \leq |K((\boldsymbol{x}_t, u_t)(\boldsymbol{x}_t, u_t))| \leq K_{\max}$, we know

$$\frac{\sigma_{u_t, t-1}^2(\boldsymbol{x}_t)}{\lambda} \leq \min\{\frac{K_{\max}}{\lambda}, \frac{\sigma_{u_t, t-1}^2(\boldsymbol{x}_t)}{\lambda}\} \leq \frac{K_{\max}}{\lambda} \min\{1, \frac{\sigma_{u_t, t-1}^2(\boldsymbol{x}_t)}{\lambda}\}$$

which leads to

$$\sum_{t=1}^T \frac{\sigma_{u_t, t-1}^2(\boldsymbol{x}_t)}{\lambda} \leq \frac{K_{\max}}{\lambda} \sum_{t=1}^T \min\{1, \frac{1}{\lambda} \sigma_{u_t, t-1}^2(\boldsymbol{x}_t)\} \leq \frac{2K_{\max}}{\lambda} \sum_{t=1}^T \log\left(1 + \frac{1}{\lambda} \sigma_{u_t, t-1}^2(\boldsymbol{x}_t)\right).$$

by applying the fact that $\min\{1, x\} \leq 2\log(1+x)$ for $x \geq 0$.

We can summarize the above two conditions for $\lambda$ together and achieve

$$\sum_{t=1}^T \sigma_{u_t, t-1}(\boldsymbol{x}_t) \leq \sqrt{2T \max\{1, K_{\max}\} \sum_{t=1}^T \log\left(1 + \frac{1}{\lambda} \sigma_{u_t, t-1}^2(\boldsymbol{x}_t)\right)}. \tag{25}$$

Now we can use the property of the Shur complement for $K_t$:

$$\det\left(\boldsymbol{I}_t + \frac{1}{\lambda}\boldsymbol{K}_t\right) = \det\left(\boldsymbol{I}_{t-1} + \frac{1}{\lambda}\boldsymbol{K}_{t-1}\right)$$
$$\times \left[1 + \frac{1}{\lambda}\big(\underbrace{K((\boldsymbol{x}_t, u_t), (\boldsymbol{x}_t, u_t)) - \boldsymbol{k}_{t-1}(\boldsymbol{x}_t, u_t)^\top (\boldsymbol{K}_{t-1} + \lambda\boldsymbol{I})^{-1}\boldsymbol{k}_{t-1}(\boldsymbol{x}_t, u_t)}_{\sigma^2_{u_t, t-1}(\boldsymbol{x}_t)}\big)\right]$$

which leads to

$$\sum_{t=1}^{T}\log\left(1 + \frac{1}{\lambda}\sigma^2_{u_t, t-1}(\boldsymbol{x}_t)\right) = \sum_{t=1}^{T}\log\frac{\det\left(\boldsymbol{I}_t + \frac{1}{\lambda}\boldsymbol{K}_t\right)}{\det\left(\boldsymbol{I}_{t-1} + \frac{1}{\lambda}\boldsymbol{K}_{t-1}\right)} = \log\det\left(\boldsymbol{I}_T + \lambda^{-1}\boldsymbol{K}_T\right).$$

Therefore, we combine above result with equation 25 and obtain

$$\sum_{t=1}^{T}\sigma_{u_t, t-1}(\boldsymbol{x}_t) \leq \sqrt{2T\max\{1, K_{\max}\}\cdot\log\det(\boldsymbol{I}_T + \lambda^{-1}\boldsymbol{K}_T)}$$

$\square$

## D.4 PROOF OF LEMMA C.5

*Proof.* We note that

$$\|\delta_t(\boldsymbol{x}, u)\|^2_{\ell^2} = \|\phi((\boldsymbol{x}, u))\|^2_{\ell^2} + \left\|\boldsymbol{\Phi}_t^\top\boldsymbol{\Sigma}_t^{-1}\boldsymbol{k}_t(\boldsymbol{x}, u)\right\|^2_{\ell^2} - 2\langle\phi((\boldsymbol{x}, u)), \boldsymbol{\Phi}_t^\top\boldsymbol{\Sigma}_t^{-1}\boldsymbol{k}_t(\boldsymbol{x}, u)\rangle_{\ell^2}$$

and we have

$$\left\|\boldsymbol{\Phi}_t^\top\boldsymbol{\Sigma}_t^{-1}\boldsymbol{k}_t(\boldsymbol{x}, u)\right\|^2_{\ell^2} = \boldsymbol{k}_t(\boldsymbol{x}, u)^\top\boldsymbol{\Sigma}_t^{-1}\boldsymbol{\Phi}_t\boldsymbol{\Phi}_t^\top\boldsymbol{\Sigma}_t^{-1}\boldsymbol{k}_t(\boldsymbol{x}, u)$$
$$= \boldsymbol{k}_t(\boldsymbol{x}, u)^\top\boldsymbol{\Sigma}_t^{-1}\boldsymbol{K}_t\boldsymbol{\Sigma}_t^{-1}\boldsymbol{k}_t(\boldsymbol{x}, u)$$
$$= \boldsymbol{k}_t(\boldsymbol{x}, u)^\top\boldsymbol{\Sigma}_t^{-1}\boldsymbol{\Sigma}_t\boldsymbol{\Sigma}_t^{-1}\boldsymbol{k}_t(\boldsymbol{x}, u) - \lambda\boldsymbol{k}_t(\boldsymbol{x}, u)^\top\boldsymbol{\Sigma}_t^{-2}\boldsymbol{k}_t(\boldsymbol{x}, u)$$
$$= \boldsymbol{k}_t(\boldsymbol{x}, u)^\top\boldsymbol{\Sigma}_t^{-1}\boldsymbol{k}_t(\boldsymbol{x}, u) - \lambda\boldsymbol{k}_t(\boldsymbol{x}, u)^\top\boldsymbol{\Sigma}_t^{-2}\boldsymbol{k}_t(\boldsymbol{x}, u)$$

and

$$\langle\phi((\boldsymbol{x}, u)), \boldsymbol{\Phi}_t^\top\boldsymbol{\Sigma}_t^{-1}\boldsymbol{k}_t(\boldsymbol{x}, u)\rangle_{\ell^2} = \phi((\boldsymbol{x}, u))^\top\boldsymbol{\Phi}_t^\top\boldsymbol{\Sigma}_t^{-1}\boldsymbol{k}_t(\boldsymbol{x}, u) = \boldsymbol{k}_t(\boldsymbol{x}, u)^\top\boldsymbol{\Sigma}_t^{-1}\boldsymbol{k}_t(\boldsymbol{x}, u).$$

Putting above equalities together, we have

$$\|\delta_t(\boldsymbol{x}, u)\|^2_{\ell^2} = \|\phi((\boldsymbol{x}, u))\|^2_{\ell^2} - \boldsymbol{k}_t(\boldsymbol{x}, u)^\top\boldsymbol{\Sigma}_t^{-1}\boldsymbol{k}_t(\boldsymbol{x}, u) - \lambda\boldsymbol{k}_t(\boldsymbol{x}, u)^\top\boldsymbol{\Sigma}_t^{-2}\boldsymbol{k}_t(\boldsymbol{x}, u)$$
$$= K((\boldsymbol{x}, u), (\boldsymbol{x}, u)) - \boldsymbol{k}_t(\boldsymbol{x}, u)^\top\boldsymbol{\Sigma}_t^{-1}\boldsymbol{k}_t(\boldsymbol{x}, u) - \lambda\boldsymbol{k}_t(\boldsymbol{x}, u)^\top\boldsymbol{\Sigma}_t^{-2}\boldsymbol{k}_t(\boldsymbol{x}, u)$$
$$= \sigma^2_{u, t}(\boldsymbol{x}) - \lambda\boldsymbol{k}_t(\boldsymbol{x}, u)^\top\boldsymbol{\Sigma}_t^{-2}\boldsymbol{k}_t(\boldsymbol{x}, u)$$
$$\leq \sigma^2_{u, t}(\boldsymbol{x})$$

since $\boldsymbol{\Sigma}_t^{-1}$ is positive semindefinite.

$\square$

## D.5 PROOF OF LEMMA C.6

*Proof.* We first define

$$\boldsymbol{s}_t = \boldsymbol{\Phi}_t\boldsymbol{\epsilon}_t = \sum_{s=1}^{t}\phi(\boldsymbol{x}_s, u_s)\epsilon_s.$$

Note that $\boldsymbol{s}_t$ is a martingale w.r.t $\mathcal{F}_t$.

Also we define a supermartingale

$$M_t(\boldsymbol{g}) = exp\big(\sum_{s=1}^{t}\frac{1}{\sigma}\langle\boldsymbol{g}, \boldsymbol{s}_t\rangle - \frac{1}{2}\|\boldsymbol{g}\|^2\big)$$

which has an alternative form

$$M_t(\boldsymbol{g}) = exp\big(\sum_{s=1}^{t} \frac{1}{\sigma}\langle \boldsymbol{g}, \phi(\boldsymbol{x}_s, u_s)\rangle \epsilon_s - \frac{1}{2}\|\boldsymbol{g}\|^2\big)$$

where $\boldsymbol{g}$ is the function vector with elements

We follow the approach from classical linear bandit Abbasi-Yadkori et al. (2011), which is averaging $M_t(\boldsymbol{g})$ w.r.t a Gaussian distribution on $\boldsymbol{g}$. The key technical issue is the infinite dimension of the function vector $\boldsymbol{g}$. We will first perform the truncated version which can precisely match the classical result. Let $d$ be the dimension of the feature map. Our target is the obtain the limiting result when $d \to \infty$. Now assume $\boldsymbol{g}^d \sim \mathcal{N}(\boldsymbol{0}, \frac{1}{\lambda}\boldsymbol{I}_d)$, independent of everything else, and define

$$M_t^{(d)} = \mathbb{E}_{\boldsymbol{g}^d}[M_t(\boldsymbol{g}^d)] = \int M_t^{(d)}(\boldsymbol{g})d\rho^d(\boldsymbol{g})$$

and by iterated expectation (i.e Fubini's theorem), we have

$$\mathbb{E}[M_t^{(d)}|\mathcal{F}_t] \leq M_{t-1}$$

which shows that $M_t$ is a supermartingale.

Then we define $\Psi : \ell^2 \to \mathbb{R}^d$ as the truncation projection onto the first $d$ coordinates: $\Psi_d\boldsymbol{\theta} = [\boldsymbol{\Theta}_1, \cdots, \boldsymbol{\Theta}_d]^\top$ for any $\boldsymbol{\theta} \in \ell^2$. We further denote

$$\boldsymbol{\Psi}_d\boldsymbol{\Phi}_t^\top = [\Psi_d\phi(\boldsymbol{x}_1, u_1), \cdots, \Psi_d\phi(\boldsymbol{x}_t, u_t)] \in \mathbb{R}^{d\times t}$$

and

$$\boldsymbol{\Psi}_d\boldsymbol{J}_t\boldsymbol{\Psi}_d^\top = \boldsymbol{\Psi}_d\boldsymbol{\Phi}_t^\top\boldsymbol{\Phi}_t\boldsymbol{\Psi}_d.$$

We notices that

$$\frac{\det(\lambda\boldsymbol{I}_d)}{\det\big(\lambda\boldsymbol{I}_d + \boldsymbol{\Psi}_d\boldsymbol{J}_t\boldsymbol{\Psi}_d^\top\big)} = \frac{1}{\det\big(\boldsymbol{I}_d + \lambda^{-1}\boldsymbol{\Psi}_d\boldsymbol{J}_t\boldsymbol{\Psi}_d^\top\big)}$$

which leads to

$$M_t^{(d)} = \Big(\frac{\det(\lambda\boldsymbol{I}_d)}{\det\big(\lambda\boldsymbol{I}_d + \boldsymbol{\Psi}_d\boldsymbol{J}_t\boldsymbol{\Psi}_d^\top\big)}\Big)^{1/2} \exp\Big(\frac{1}{2\sigma^2}\|\boldsymbol{\Psi}_d\boldsymbol{\Phi}_t\boldsymbol{\epsilon}_t\|_{(\lambda\boldsymbol{I}_d + \boldsymbol{\Psi}_d\boldsymbol{J}_t\boldsymbol{\Psi}_d^\top)^{-1}}^2\Big)$$

$$= \det\big(\boldsymbol{I}_d + \lambda^{-1}\boldsymbol{\Psi}_d\boldsymbol{J}_t\boldsymbol{\Psi}_d^\top\big)^{-1/2} \exp\Big(\frac{1}{2\sigma^2}\|\boldsymbol{\Psi}_d\boldsymbol{\Phi}_t\boldsymbol{\epsilon}_t\|_{(\lambda\boldsymbol{I}_d + \boldsymbol{\Psi}_d\boldsymbol{J}_t\boldsymbol{\Psi}_d^\top)^{-1}}^2\Big).$$

Let $M_t$ be the limit of $M_t^{(d)}$ as $d \to \infty$, we have

$$M_t = \det\big(\boldsymbol{I}_\infty + \lambda^{-1}\boldsymbol{J}_t\big)^{-1/2} \exp\Big(\frac{1}{2\sigma^2}\|\boldsymbol{\Phi}_t\boldsymbol{\epsilon}_t\|_{(\lambda\boldsymbol{I}_\infty + \boldsymbol{J}_t)^{-1}}^2\Big)$$

$$= \det\big(\boldsymbol{I}_t + \lambda^{-1}\boldsymbol{K}_t\big)^{-1/2} \exp\Big(\frac{1}{2\sigma^2}\|\boldsymbol{\Phi}_t\boldsymbol{\epsilon}_t\|_{\Gamma_t^{-1}}^2\Big)$$

where the second step is from (Slyvestr) or Weinstein–Aronszajn identity.

By Ville's inequality,

$$\mathbb{P}\big(\sup_{t=0,1,2,\cdots} M_t \geq \frac{1}{\delta}\big) \leq \mathbb{E}[M_0] \cdot \delta$$

and $M_0 = 1$. Thus we know that, with probability at least $1 - \delta$, for all $t = 0, 1, 2, \cdots$

$$\log(M_t) \leq \log\Big(\frac{1}{\delta}\Big)$$

which leads to

$$-\frac{1}{2}\log\det\big(\boldsymbol{I}_t + \lambda^{-1}\boldsymbol{K}_t\big) + \frac{1}{2\sigma^2}\|\boldsymbol{\Phi}_t\boldsymbol{\epsilon}_t\|_{\Gamma_t^{-1}}^2 \leq \log\Big(\frac{1}{\delta}\Big).$$

After re-arranging, we get

$$\|\boldsymbol{\Phi}_t\boldsymbol{\epsilon}_t\|_{\Gamma_t^{-1}}^2 \leq 2\sigma^2 \log\frac{\sqrt{\det(\boldsymbol{I}_t + \lambda^{-1}\boldsymbol{K}_t)}}{\delta}.$$

which shows our result.

$\square$

## D.6 PROOF OF LEMMA C.4

*Proof.* Let us write $\boldsymbol{\Phi}_t = \boldsymbol{U}_t \boldsymbol{\Lambda}_t \boldsymbol{V}_t^\top$ as the singular value decomposition(SVD) of $\boldsymbol{\Phi}_t$. We have $\boldsymbol{\Lambda}_t = [\boldsymbol{\Lambda}_{1,t}, \boldsymbol{0}]$ where $\boldsymbol{\Lambda}_{1,t}$ is a $t \times t$ diagonal matrix with singular values of $\boldsymbol{\Phi}_t$. We also note that $\boldsymbol{\Sigma}_t \in \mathbb{R}^{t \times t}$ and $\boldsymbol{U}_t \in \mathbb{R}^{t \times t}$. We also have

$$\boldsymbol{J}_t = \boldsymbol{\Phi}_t^\top \boldsymbol{\Phi}_t = \boldsymbol{V}_t \begin{bmatrix} \boldsymbol{\Lambda}_{1,t}^2 & \boldsymbol{0} \\ \boldsymbol{0} & \boldsymbol{0} \end{bmatrix} \boldsymbol{V}_t^\top$$

and similarly

$$\boldsymbol{K}_t = \boldsymbol{\Phi}_t \boldsymbol{\Phi}_t^\top = \boldsymbol{U}_t \boldsymbol{\Lambda}_{1,t}^2 \boldsymbol{U}_t^\top.$$

Then, we have

$$\boldsymbol{\Gamma}_t = \boldsymbol{V}_t \begin{bmatrix} \boldsymbol{\Lambda}_{1,t}^2 + \lambda \boldsymbol{I}_t & \boldsymbol{0} \\ \boldsymbol{0} & \lambda \boldsymbol{I}_\infty \end{bmatrix} \boldsymbol{V}_t^\top, \quad \boldsymbol{\Sigma}_t = \boldsymbol{U}_t(\boldsymbol{\Lambda}_{1,t}^2 + \lambda \boldsymbol{I}_t)\boldsymbol{U}_t^\top.$$

It is clear to have the identity:

$$\boldsymbol{\Sigma}_t^{-1}\boldsymbol{\Phi}_t = \boldsymbol{\Phi}_t \boldsymbol{\Gamma}_t^{-1}$$

since both side equal $\boldsymbol{U}_t[\boldsymbol{D}_t, \boldsymbol{0}]\boldsymbol{V}_t^\top$ where $\boldsymbol{D}_t = \boldsymbol{\Lambda}_{1,t}(\boldsymbol{\Lambda}_{1,t}^2 + \lambda \boldsymbol{I}_t)^{-1}$, which is a diagonal matrix.

Next, we note that

$$\begin{aligned} \sigma_{u,t}^2(\boldsymbol{x}) &= K((\boldsymbol{x},u),(\boldsymbol{x},u)) - \boldsymbol{k}_t(\boldsymbol{x},u)^\top \boldsymbol{\Sigma}_t^{-1} \boldsymbol{k}_t(\boldsymbol{x},u) \\ &= \phi(\boldsymbol{x},u)^\top (\boldsymbol{I}_\infty - \boldsymbol{\Phi}_t^\top \boldsymbol{\Sigma}_t^{-1} \boldsymbol{\Phi}_t)\phi(\boldsymbol{x},u) \\ &= \phi(\boldsymbol{x},u)^\top (\boldsymbol{I}_\infty - \boldsymbol{\Phi}_t^\top \boldsymbol{\Phi}_t \boldsymbol{\Gamma}_t^{-1})\phi(\boldsymbol{x},u) \end{aligned}$$

which is a norm of $\phi(\boldsymbol{x},u)$ induced by matrix

$$\begin{aligned} \boldsymbol{I}_\infty - \boldsymbol{\Phi}_t^\top \boldsymbol{\Phi}_t \boldsymbol{\Gamma}_t^{-1} &= \boldsymbol{I}_\infty - \boldsymbol{J}_t \boldsymbol{\Gamma}_t^{-1} \\ &= \boldsymbol{V}_t \begin{bmatrix} \lambda \boldsymbol{I}_t(\boldsymbol{\Lambda}_{1,t}^2 + \lambda \boldsymbol{I}_t)^{-1} & \boldsymbol{0} \\ \boldsymbol{0} & \lambda \boldsymbol{I}_\infty \end{bmatrix} \boldsymbol{V}_t^\top \\ &= \lambda \boldsymbol{V}_t \begin{bmatrix} (\boldsymbol{\Lambda}_{1,t}^2 + \lambda \boldsymbol{I}_t)^{-1} & \boldsymbol{0} \\ \boldsymbol{0} & \boldsymbol{I}_\infty \end{bmatrix} \boldsymbol{V}_t^\top \\ &= \lambda \boldsymbol{\Gamma}_t^{-1}. \end{aligned}$$

Therefore, we have the other identity

$$\sigma_{u,t}^2(\boldsymbol{x}) = \lambda \|\phi(\boldsymbol{x},u)\|_{\boldsymbol{\Gamma}_t^{-1}}^2.$$

$\square$

## E MISCELLANEOUS

### E.1 ALGORITHMS

---
**Algorithm 1** LK-GP-UCB
---
1: **Input:** $T$, $\lambda$, $\{\beta_t\}_{t=1}^T$
2: **Initialization:** $\mu_{u,0}(\boldsymbol{x})$, $\sigma_{u,0}(\boldsymbol{x})$
3: **for** $t = 1, ..., T$ **do**
4:     Observe user $u_t$ and arm set $\mathcal{D}_t$.
5:     Select arm $\boldsymbol{x}_t = \arg\max_{\boldsymbol{x} \in \mathcal{D}_t} \mu_{u_t,t-1}(\boldsymbol{x}) + \beta_t \sigma_{u_t,t-1}(\boldsymbol{x})$.
6:     Receive feedback $y_t = f(\boldsymbol{x}_t, u_t) + \epsilon_t$.
7:     Update $\mu_{u_t,t}(\boldsymbol{x})$ and $\sigma_{u_t,t}^2(\boldsymbol{x})$.
8: **end for**
---

---

**Algorithm 2** `LK-GP-TS`

---

1: **Input:** $T$, $\lambda$, $\{\nu_t\}_{t=1}^T$
2: **Initialization:** $\mu_{u,0}(\boldsymbol{x})$, $\sigma_{u,0}(\boldsymbol{x})$
3: **for** $t = 1, ..., T$ **do**
4:      Observe user $u_t$ and arm set $\mathcal{D}_t$.
5:      Sample $\widetilde{\mu}_t(\boldsymbol{x})$ from $\mathcal{N}(\mu_{u_t,t-1}(\boldsymbol{x}), \nu_t^2 \sigma_{u_t,t-1}^2(\boldsymbol{x}))$ for all $\boldsymbol{x} \in \mathcal{D}_t$
6:      Select arm $\boldsymbol{x}_t = \arg\max_{\boldsymbol{x} \in \mathcal{D}_t} \widetilde{\mu}_t(\boldsymbol{x})$.
7:      Receive feedback $y_t = f(\boldsymbol{x}_t, u_t) + \epsilon_t$.
8:      Update $\mu_{u_t,t}(\boldsymbol{x})$ and $\sigma_{u_t,t}^2(\boldsymbol{x})$.
9: **end for**

---

### E.2   RECURSIVE UPDATE OF POSTERIOR MEAN AND VARIANCE

This sections refers to the derivation of incremental update of the posterior mean and posterior variance Chowdhury & Gopalan (2017), via the properties of Schur complement. Recall that we need to handle the inversion of $\boldsymbol{\Sigma}_t = \boldsymbol{I} + \lambda \boldsymbol{K}_t \in \mathbb{R}^{t \times t}$ which grows with the number of rounds. To compute the inversion of $\boldsymbol{\Sigma}_t$ efficiently, we use the recursive formula from $\boldsymbol{\Sigma}_{t-1}$ by block matrix inverse formula

$$\boldsymbol{\Sigma}_t^{-1} = \begin{bmatrix} \boldsymbol{M}_{11,t} & \boldsymbol{M}_{12,t} \\ \boldsymbol{M}_{12,t}^\top & d_t^{-1} \end{bmatrix} \tag{26}$$

where

$$\begin{aligned} \boldsymbol{M}_{11,t} &= \boldsymbol{\Sigma}_{t-1}^{-1} + d_t^{-1} \boldsymbol{G}_t \\ \boldsymbol{M}_{12,t} &= -d_t^{-1} \boldsymbol{\Sigma}_{t-1}^{-1} \boldsymbol{k}_{t-1}(\boldsymbol{x}_t, u_t) \end{aligned} \tag{27}$$

and

$$d_t = K((\boldsymbol{x}_t, u_t), (\boldsymbol{x}_t, u_t)) - \boldsymbol{k}_{t-1}(\boldsymbol{x}_t, u_t)^\top \boldsymbol{\Sigma}_{t-1}^{-1} \boldsymbol{k}_{t-1}(\boldsymbol{x}_t, u_t) + \lambda = \sigma_{u_t,t-1}^2(\boldsymbol{x}_t) + \lambda$$
$$\boldsymbol{G}_t = \boldsymbol{\Sigma}_{t-1}^{-1} \boldsymbol{k}_{t-1}(\boldsymbol{x}_t, u_t) \boldsymbol{k}_{t-1}(\boldsymbol{x}_t, u_t)^\top \boldsymbol{\Sigma}_{t-1}^{-1}$$

Here $d_t$ is the Schur complement.

Thus we have the posterior mean using equation 26

$$\begin{aligned} \mu_{u,t}(\boldsymbol{x}) &= \boldsymbol{k}_t(\boldsymbol{x}, u)^\top \boldsymbol{\Sigma}_t^{-1} \boldsymbol{y}_t \\ &= \begin{bmatrix} \boldsymbol{k}_{t-1}(\boldsymbol{x}, u)^\top & K((\boldsymbol{x}, u), (\boldsymbol{x}_t, u_t)) \end{bmatrix} \begin{bmatrix} \boldsymbol{M}_{11,t} & \boldsymbol{M}_{12,t} \\ \boldsymbol{M}_{12,t}^\top & d_t^{-1} \end{bmatrix} \begin{bmatrix} \boldsymbol{y}_{t-1} \\ y_t \end{bmatrix} \\ &= \boldsymbol{k}_{t-1}(\boldsymbol{x}, u)^\top \boldsymbol{M}_{11,t} \boldsymbol{y}_{t-1} + K((\boldsymbol{x}, u), (\boldsymbol{x}_t, u_t)) \boldsymbol{M}_{12,t}^\top \boldsymbol{y}_{t-1} \\ &\quad + \boldsymbol{k}_{t-1}(\boldsymbol{x}, u)^\top \boldsymbol{M}_{12,t} y_t + K((\boldsymbol{x}, u), (\boldsymbol{x}_t, u_t)) d_t^{-1} y_t \\ &= \underbrace{\boldsymbol{k}_{t-1}(\boldsymbol{x}, u)^\top \boldsymbol{\Sigma}_{t-1}^{-1} \boldsymbol{y}_{t-1}}_{\mu_{u,t-1}(\boldsymbol{x})} + d_t^{-1}(\beta_1 \boldsymbol{y}_{t-1} - \beta_2 \boldsymbol{y}_{t-1} - \beta_3 y_t + \beta_4 y_t) \end{aligned}$$

where

$$\beta_1 = \boldsymbol{k}_{t-1}(\boldsymbol{x}, u)^\top \boldsymbol{G}_t \Rightarrow \beta_1 \boldsymbol{y}_{t-1} = \left( \boldsymbol{k}_{t-1}(\boldsymbol{x}, u)^\top \boldsymbol{\Sigma}_{t-1}^{-1} \boldsymbol{k}_{t-1}(\boldsymbol{x}_t, u_t) \right) \mu_{u_t,t-1}(\boldsymbol{x}_t)$$
$$\beta_2 = K((\boldsymbol{x}, u), (\boldsymbol{x}_t, u_t)) \boldsymbol{k}_{t-1}(\boldsymbol{x}_t, u_t)^\top \boldsymbol{\Sigma}_{t-1}^{-1} \Rightarrow \beta_2 \boldsymbol{y}_{t-1} = K((\boldsymbol{x}, u), (\boldsymbol{x}_t, u_t)) \mu_{u_t,t-1}(\boldsymbol{x}_t)$$
$$\beta_3 = \boldsymbol{k}_{t-1}(\boldsymbol{x}, u)^\top \boldsymbol{\Sigma}_{t-1}^{-1} \boldsymbol{k}_{t-1}$$
$$\beta_4 = K((\boldsymbol{x}, u), (\boldsymbol{x}_t, u_t)).$$

Thus we have the recursive update of posterior mean

$$\begin{aligned} \mu_{u,t}(\boldsymbol{x}) &= \mu_{u,t-1}(\boldsymbol{x}) + d_t^{-1} \\ &\quad \times \left( \boldsymbol{k}_{t-1}(\boldsymbol{x}, u)^\top \boldsymbol{\Sigma}_{t-1}^{-1} \boldsymbol{k}_{t-1}(\boldsymbol{x}_t, u_t)(\mu_{u_t,t-1}(\boldsymbol{x}_t) - y_t) + K((\boldsymbol{x}, u), (\boldsymbol{x}_t, u_t))(y_t - \mu_{u_t,t-1}(\boldsymbol{x}_t)) \right) \\ &= \mu_{u,t-1}(\boldsymbol{x}) + d_t^{-1} q_{t-1}((\boldsymbol{x}, u), (\boldsymbol{x}_t, u_t))(y_t - \mu_{u_t,t-1}(\boldsymbol{x}_t)) \end{aligned}$$

where $q_{t-1}((\boldsymbol{x}, u), (\boldsymbol{x}_t, u_t))$ is defined from

$$q_t((\boldsymbol{x}, u), (\boldsymbol{x}', u')) = K((\boldsymbol{x}, u), (\boldsymbol{x}', u')) - \boldsymbol{k}_t(\boldsymbol{x}, u)^\top \boldsymbol{\Sigma}_t^{-1} \boldsymbol{k}_t(\boldsymbol{x}', u')$$

which can be transferred into a recursive form using equation 26

$$q_t((\boldsymbol{x}, u), (\boldsymbol{x}', u'))$$

$$=K((\boldsymbol{x}, u), (\boldsymbol{x}', u')) - \Big( \boldsymbol{k}_{t-1}(\boldsymbol{x}, u)^\top \boldsymbol{\Sigma}_{t-1}^{-1} \boldsymbol{k}_{t-1}(\boldsymbol{x}', u')$$

$$+ d_t^{-1}(\beta_1 \boldsymbol{k}_{t-1}(\boldsymbol{x}', u') - \beta_2 \boldsymbol{k}_{t-1}(\boldsymbol{x}', u') - \beta_3 K((\boldsymbol{x}_t, u_T), (\boldsymbol{x}', u')) + \beta_4 K((\boldsymbol{x}_t, u_T), (\boldsymbol{x}', u'))) \Big)$$

$$=q_{t-1}((\boldsymbol{x}, u), (\boldsymbol{x}', u')) - d_t^{-1} q_{t-1}((\boldsymbol{x}, u), (\boldsymbol{x}_t, u_t)) q_{t-1}((\boldsymbol{x}_t, u_t), (\boldsymbol{x}', u')).$$

Now using the incremental update of the posterior covariance, we can easily obtain the recursive update for the posterior variance

$$\sigma_{u,t}^2(\boldsymbol{x}) = \sigma_{u,t-1}^2(\boldsymbol{x}) - d_t^{-1} q_{t-1}^2((\boldsymbol{x}, u), (\boldsymbol{x}_t, u_t)).$$

Now replace $d_t$ by $\sigma_{u_t,t-1}^2(\boldsymbol{x}_t) + \lambda$ and we achieve the recursive updates in equation 8.

# F    SUPPLEMENT TO EXPERIMENTS

This appendix provides full details of our synthetic environments, algorithm configurations, hyper-parameter selection, implementation choices, ablations, and reporting protocol.

## F.1    SYNTHETIC ENVIRONMENTS

Let $\mathcal{U} = \{1, \ldots, n\}$ denote users, $\mathcal{D} \subset \mathbb{R}^d$ the arm (context) space, and $M_t := |\mathcal{D}_t|$ the number of candidates shown at round $t$. We draw a global normalized context pool $\mathcal{D} = \{\boldsymbol{x}^{(1)}, \ldots, \boldsymbol{x}^{(m)}\}$ with $\boldsymbol{x}^{(i)} \sim \mathcal{N}(\boldsymbol{0}, \boldsymbol{I}_d)$ and $\boldsymbol{x}^{(i)} \leftarrow \boldsymbol{x}^{(i)}/\|\boldsymbol{x}^{(i)}\|$. At round $t$ we present $\mathcal{D}_t$ by sampling $M_t$ distinct items from $\mathcal{D}$ without replacement. One user $u_t$ is served per round, drawn uniformly from $\mathcal{U}$ unless stated otherwise. Rewards are observed with additive noise $y_t = f(\boldsymbol{x}_t, u_t) + \epsilon_t$. We generate graphs, contexts, and ground-truth rewards under one linear regime (*Linear–GOB*) and two kernelized regimes (*Laplacian–Kernel* using GP draw and representer draw).

**User graph.** We consider two graph random generators on $\mathcal{U}$. First random graph family is Erdős–Rényi (ER) random graphs: each (undirected) edge is present with probability $p$ and weights $w_{ij} = 1$. We set $p = 0.2$ in our experiment. Second one is Radial basis function(RBF) random graphs: sample latent $\boldsymbol{z}_i \sim \mathcal{N}(\boldsymbol{0}, \boldsymbol{I}_q)$, set $w_{ij} = \exp(-\rho_L \|\boldsymbol{z}_i - \boldsymbol{z}_j\|_2^2)$, and sparsify by keeping edges with $w_{ij} \geq s$. We choose $s = 0.1$, $\rho_L = 0.1$ and $q = 4$ in our simulation.

**Task Design.** We design different level of the task. The simplest case is $(M, M_t, n, d, T) = (10, 5, 20, 5, 1000)$. This is a 10-arm bandit problem with $50\%$ viewability at each round for all users. The medium level is $(M, M_t, n, d, T) = (20, 5, 20, 10, 3000)$ which leads to a 20-arm bandit problem with $25\%$ viewability at each round for all users. We also have the toughest case using $(M, M_t, n, d, T) = (50, 5, 20, 20, 3000)$ which leads to a 50-arm bandit problem with $10\%$ viewability at each round for all users. $\sigma$ is set as $0.1$ unless additional specification.

**Practical scenarios.** Although our empirical study uses synthetic environments, the multi-user, graph-based bandit setting we consider is motivated by several practical applications. Examples include recommendation systems, where users are connected via social or similarity graphs and repeatedly interact with a common catalog of items; regional personalization problems, where stores or geographic areas form a graph and the arms correspond to assortments or pricing actions; and applications in healthcare or education, where patients or students are linked through similarity networks while treatments or exercises constitute the arm set. In such domains, the proposed Laplacian kernelized bandits can leverage the user graph to share statistical strength while capturing non-linear context effects.

### F.1.1    REGIME 1: *Linear–GOB* (GRAPH-SMOOTH LINEAR REWARDS)

Sample initial user parameters $\boldsymbol{\Theta}_0 \in \mathbb{R}^{n \times d}$ with rows $\boldsymbol{\theta}_{0,i} \sim \mathcal{N}(\boldsymbol{0}, \boldsymbol{I}_d)$. Enforce the graph homophily via Tikhonov smoothingYankelevsky & Elad (2016):

$$\boldsymbol{\Theta} = \arg\min_{\tilde{\boldsymbol{\Theta}}} \|\tilde{\boldsymbol{\Theta}} - \boldsymbol{\Theta}_0\|_F^2 + \eta \operatorname{tr}(\tilde{\boldsymbol{\Theta}}^\top \boldsymbol{L} \tilde{\boldsymbol{\Theta}}) = (\boldsymbol{I}_n + \eta \boldsymbol{L})^{-1} \boldsymbol{\Theta}_0.$$

Thus $f(\boldsymbol{x}, u) = \boldsymbol{x}^\top \boldsymbol{\theta}_u$, where $\boldsymbol{\theta}_u$ is row $u$ of $\boldsymbol{\Theta}$. The strength of the graph homophily $\eta$ is set as 1.0 as default.

### F.1.2 REGIME 2: *Laplacian-Kernel*

Our choice of the base kernel $K_x$ over arms is *Squared Exponential* which are defined as

$$K_{\mathrm{SE}}(\boldsymbol{x}, \boldsymbol{x}') = \exp\left(-\|\boldsymbol{x} - \boldsymbol{x}'\|^2 / 2\ell^2\right)$$

where length-scale $\ell > 0$ and is set to be 1.0 in our experiment. Then we construct the *multi-user kernel* by the definition:

$$K((\boldsymbol{x}, u), (\boldsymbol{x}', u')) \;=\; [\boldsymbol{L}_\rho^{-1/2}]_{u,u'} \, K_x(\boldsymbol{x}, \boldsymbol{x}')$$

where we set $\rho = 0.01$ in our experiment.

**Option A: *Laplacian-Kernel* with GP draw**

We draw the joint values $\{f(\boldsymbol{x}, u)\}_{u \in \mathcal{U}, \boldsymbol{x} \in \mathcal{D}}$ from the zero-mean GP with covariance induced by $K$ and fix $f$ by interpolation on $\mathcal{D} \times \mathcal{U}$. Noise is $\epsilon_t \sim \mathcal{N}(0, \sigma^2)$ with $\sigma = 0.01 \cdot \mathrm{range}(f)$.

**Option B: *Laplacian-Kernel* with representer draw** We consider the representer theorem for RKHS and sample the i.i.d. coefficients via $\alpha_{\boldsymbol{x}, u} \sim \mathcal{N}(0, 1)$ on $\mathcal{D} \times \mathcal{U}$ and set

$$f(\boldsymbol{x}, u) \;=\; \sum_{u' \in \mathcal{U}, \boldsymbol{x}' \in \mathcal{D}} \alpha_{\boldsymbol{x}, u} \, K\big((\boldsymbol{x}, u), (\boldsymbol{x}', u')\big).$$

### F.2 BASELINES

All methods face the *same* sequence $\{u_t, \mathcal{D}_t, \epsilon_t\}_{t=1}^T$ in each trial of each synthetic environment to ensure a fair comparison. Our experiment include the following baselines.

**Per-User LinUCB(no graph).**: We implement `Per-User LinUCB`, which ignores the whole graph and perform the linear bandit algorithm independently on each user.

**Pooled LinUCB(no graph).**: We implement `Pooled LinUCB`, which ignores graph and personalization by treating the multi-user problem as a single agent bandit problem. Simply speaking, there is global linear UCB algorithm to solve the problem.

**GP-UCB(no graph).** We implement `GP-UCB`Chowdhury & Gopalan (2017), which is the `IGP-UCB` from the previous study on GP and UCB Chowdhury & Gopalan (2017). This is a kernelized baseline using $K_x$ on arms only, ignoring the similarities across users (the Laplacian).

**GoB.Lin.** We implement `GoB.Lin`, which is the classical methods in gang-og-bandits problem Cesa-Bianchi et al. (2013). This is a Laplacian-regularized linear UCB algorithm on graph-whitened features (equivalent to `GraphUCB` with $\rho = 1$ i.e $\boldsymbol{A} = \boldsymbol{I} + \boldsymbol{L}$). The confidence scale in the algorithm is tuned from the table.

**GraphUCB.** We implement `GraphUCB`Yang et al. (2020), the Laplacian-regularized LinUCB. Also, the confidence scale in the algorithm is tuned from the table.

**COOP-KernelUCB.** We implement `COOP-KernelUCB`Dubey et al. (2020), which utilizes the product kernel over agents $\times$ arms. Here we borrow the notations from their work. We consider five choices of $K_z$ (presented below); the full kernel is $K = K_z \otimes K_x$ and we apply the same UCB rule in `LK-GP-UCB`.

The five PSD options for the agent kernel $K_z$:

1. **laplacian_inv**: $\boldsymbol{K}_z \;=\; (\boldsymbol{L} + \rho \boldsymbol{I})^{-1}, \qquad \rho > 0.$
2. **heat**: $\boldsymbol{K}_z = \exp(-\tau \boldsymbol{L})$ via the spectral decomposition of $L$.
3. **spectral_rbf**: embed nodes using the $k$ lowest nontrivial Laplacian eigenvectors $Z \in \mathbb{R}^{n \times k}$ and set
$$K_z[u, u'] \;=\; \exp\left(-\frac{\|Z_u - Z_{u'}\|^2}{2\sigma_z^2}\right).$$

4. **all ones**: full cooperation, $K_z = \mathbf{1}\mathbf{1}^\top$.

5. **learned_mmd** (*network contexts*, faithful to Dubey et al. (2020)): define per-user kernel mean embeddings $\Psi_u$ of the observed arm-context distribution in $\mathcal{H}_x$, and let

$$K_{z,t}(u, u') \;=\; \exp\Big(-\frac{\|\hat{\Psi}_u(t) - \hat{\Psi}_{u'}(t)\|^2}{2\sigma_z^2}\Big),$$

where $\hat{\Psi}_u(t)$ is the empirical mean embedding of contexts observed for user $u$ up to time $t$. In our implementation we use an efficient random Fourier feature approximation for $K_x$ and update $K_{z,t}$ on a fixed schedule; users with fewer than a small threshold of observations cooperate only with themselves (diagonal entries).

For time-varying $K_z$ (learned_mmd), the GP state is rebuilt at $K_z$ refresh points using the algorithm's own history, ensuring consistency of the Gram matrix with the current kernel. FIgure shows the comparison of the choice of $K_z$ for `COOP-KernelUCB`.

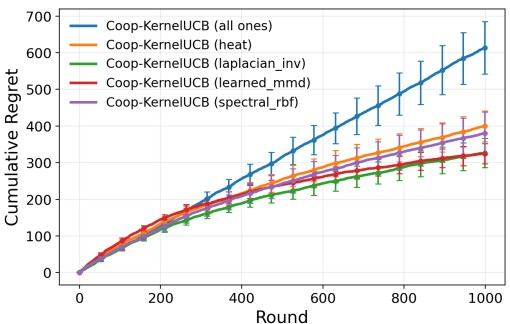

Figure 5: Comparison of the choice of user-similarity kernel for `COOP-KernelUCB`.

### F.3 CENTRALIZED PROTOCOL

At each $t$: sample $u_t \sim \mathrm{Unif}(\mathcal{U})$, present $\mathcal{D}_t$ (size $M_t$), select $\boldsymbol{x}_t \in \mathcal{D}_t$ per the algorithm, observe $y_t$, update our decision policy(model), and record $\Delta_t = \max_{\boldsymbol{x} \in \mathcal{D}_t} f(\boldsymbol{x}, u_t) - f(\boldsymbol{x}_t, u_t)$. Each configuration is repeated for $R$ trials (final results use $R = 20$; preliminary/pilot tuning uses $R \in [5, 10]$).

### F.4 POSTERIOR UPDATES AND NUMERICAL DETAILS

**Motivation.** For the original update equation 3 at round $t$, the inversion takes $\mathcal{O}(t^3|\mathcal{D}_t|)$ time. The practical updates is efficient for each pair $(\boldsymbol{x}, u)$ while it requires the updates for all pairs, leading to $\mathcal{O}(|\mathcal{D}_t|n)$ time. Therefore, high-level idea is to perform original updates equation 3 when $t \le n^{1/3}$ and perform practical updates equation 8 when when $t \le n^{1/3}$. Therefore, for our GP-based methods we use a hybrid implementation, which is described as below.

**Exact (Cholesky) phase:** maintain $\boldsymbol{\Sigma}_t = K_t + \lambda\boldsymbol{I}$ and update via rank-one Cholesky for $t < t_*$ (cost $O(t^2)$ per step; initial inversion $O(t^3)$).

**Recursive phase:** switch to the rank-one recursions in equation 8, with $q_0 = K$ restricted to $\mathcal{D} \times \mathcal{U}$. This costs $O(n|\mathcal{D}_t|)$ per update when applied to the whole grid $\mathcal{D} \times \mathcal{U}$.

By default we take $t_\star = \min\{1500, \lfloor n^{1/3}\rfloor|\mathcal{D}|\}$ as the phase switch. We use Cholesky jitter $10^{-8}$, clip negative variances to zero, and cache $K_x(\mathcal{D}, \mathcal{D})$. For large $n$ we optionally apply graph spectral truncation $\boldsymbol{L}_\rho \approx \boldsymbol{U}_r\Lambda_r\boldsymbol{U}_r^\top$ (top-$r$ eigenpairs), yielding $K \approx (\boldsymbol{U}_r\Lambda_r^{-1}\boldsymbol{U}_r^\top) \otimes K_x$.

### F.5 HYPERPARAMETERS AND TUNING

**What is fixed across algorithms.** For fairness, *base-kernel* hyperparameters are fixed inside each environment: the length-scale $\ell$ uses the median heuristic on $\mathcal{D}$, and the Laplacian ridge $\rho = 0.1$ is fixed. For $K_z$, *laplacian_inv* uses $\rho = 0.1$; *heat* uses $\tau = 1.0$; *spectral_rbf* uses $k = 8$ and median bandwidth; *learned_mmd* uses random-feature dimension 256, a median bandwidth heuristic, update

interval around 200 rounds, and a minimum count of 5 observations before a user participates in cooperation.

**What is design.** To avoid using unknown noise scale as a prior, all GP-style methods use a graph- and time-aware ridge schedule

$$\lambda_t \;=\; \lambda_{\text{base}} \cdot S_{\text{spec}} \cdot \frac{T}{T+t}, \qquad S_{\text{spec}} \;=\; \frac{\lambda_2(\boldsymbol{L})}{\lambda_{\max}(\boldsymbol{L})} \in [0,1],$$

where $\lambda_{Fiedler}(\boldsymbol{L})$ is the Fiedler value (smallest non-zero eigenvalue). We clip $\lambda_t$ to $[\lambda_{\min}, \lambda_{\max}]$ with $\lambda_{\min} = 10^{-6}$ and $\lambda_{\max} = 10^{-1}$. To limit refactorizations, we update $\lambda$ on a doubling epoch schedule (approximately at $t \approx 200, 400, 800, \ldots$) and only rebuild if the change exceeds $20\%$.

**What is tuned.** Only the exploration scales are tuned by grid search on a pilot horizon ($T_{\text{pilot}} = 1500$ for medium/hard; $T_{\text{pilot}} = 1000$ for simple) using $R_{\text{pilot}} \in \{5, 10\}$:

| Algorithm | Grid (pilot) |
|---|---|
| `LK-GP-UCB`, `GP-UCB`, `Coop-KernelUCB` | $\beta \in \{0.5, 1, 2, 4\}$ |
| `LK-GP-TS` | $\nu \in \{0.5, 1, 2, 4\}$ |
| `GOB.Lin`, `GraphUCB`, `LinUCB` variants | $\alpha \in \{0.5, 1, 2, 4\}$ |

The best pilot setting (by mean pilot cumulative regret) is then *frozen* for the full-horizon evaluation. Noise/ridge $\lambda_{\text{base}}$ in GP updates uses $\lambda_{\text{base}} \in \{0.001, 0.005, 0.01, 0.05, 0.1\}$ on the pilot.

### F.6 Ablations and Stress Tests

We report two ablation studies. One is an ablation under the *medium, Laplacian-Kernel with GP Draw* environment (ER graph, fixed $\ell$ and $\rho$) on **Scalability in users** ($n$): $n \in \{20, 50, 100, 200\}$ with fixed $(M, M_t, d, T)$ and graph generator. We provide the final cumulative regret vs. $n$ an report the last step cumulative regret in Table 1. Another study is on the effect of random graph models. Our standard experiment uses two graph random generators: Erdős–Rényi (ER) random graphs and the Radial basis function(RBF) random graphs, mentioned in F.1. We add the stochastic block models(SBM) in this ablation study. We still keep the *medium, Laplacian-Kernel with GP Draw* environment. The result is shown in Figure 6.

Table 1: Ablation over number of users $n$ (final cumulative regret; mean±SE).

| Algorithm | $n = 20$ | $n = 50$ | $n = 100$ | $n = 200$ |
|---|---|---|---|---|
| LK-GP-UCB | $627.22 \pm 32.98$ | $892.43 \pm 21.73$ | $1062.69 \pm 18.29$ | $1157.74 \pm 23.02$ |
| LK-GP-TS | $634.46 \pm 22.78$ | $943.41 \pm 19.56$ | $1176.23 \pm 15.77$ | $1260.35 \pm 16.23$ |
| Coop-KernelUCB | $730.06 \pm 31.02$ | $1015.35 \pm 22.18$ | $1273.28 \pm 17.36$ | $1358.48 \pm 14.22$ |
| GOB.Lin | $1092.86 \pm 71.70$ | $1203.32 \pm 18.57$ | $1370.51 \pm 16.78$ | $1432.48 \pm 18.72$ |
| GraphUCB | $1105.20 \pm 68.54$ | $1192.30 \pm 22.12$ | $1360.02 \pm 15.32$ | $1453.21 \pm 17.81$ |
| GP-UCB | $2222.20 \pm 90.26$ | $1964.65 \pm 61.40$ | $1641.43 \pm 37.43$ | $1444.83 \pm 36.33$ |
| Pooled-LinUCB | $2360.95 \pm 70.55$ | $1909.81 \pm 49.49$ | $1723.27 \pm 40.23$ | $1438.74 \pm 26.44$ |
| PerUser-LinUCB | $1117.87 \pm 72.04$ | $1221.99 \pm 22.03$ | $1432.89 \pm 18.81$ | $1527.04 \pm 17.61$ |

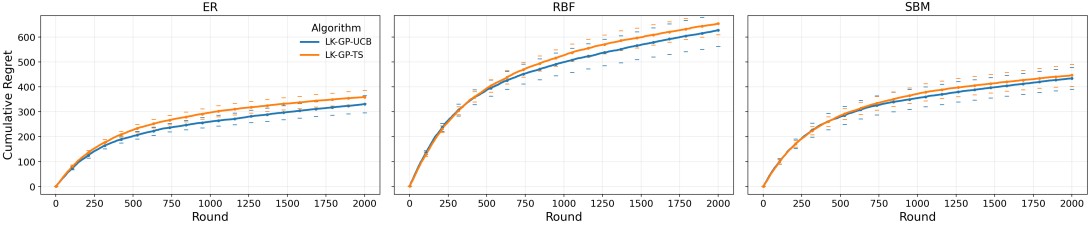

Figure 6: Comparison of the choice of random graph models.

