# OpenReview forum: "Laplacian Kernelized Bandit"
_ICLR.cc/2026/Conference — ICLR 2026 Poster_

### Official Review · Reviewer_uAXP · 2025-10-29

**Soundness:** 3
**Presentation:** 3
**Contribution:** 2
**Rating:** 4
**Confidence:** 3

**Summary:**

The paper studies multi-user contextual bandits where users are connected by a known graph and have nonlinear reward functions. It builds a joint RKHS kernel combining (i) graph smoothness via the Laplacian and (ii) nonlinear structure over arms, and then runs GP-UCB / GP-TS on this kernel. It proves regret bounds that depend on an effective dimension of the graph+kernel, not just number of users, and shows strong performance in experiments.

**Strengths:**

- Gives a clean unified kernel that fuses graph structure and nonlinear rewards in one theory.
- Provides GP-UCB / GP-TS algorithms that share information across similar users automatically.
- Regret scales with an effective dimension (can be much smaller than #users).
- Shows consistent empirical gains, especially on nonlinear graph-smooth tasks.

**Weaknesses:**

- The paper omits key related work. The problem setting is very close to prior work on graph-structured / collaborative contextual bandits [1][2], but these are not adequately discussed in the introduction or positioned against the proposed approach.

- The baseline comparison is incomplete. Modern neural contextual bandits can learn richer representations than fixed kernels, e.g. neural GP / neural UCB style methods [3]. These should be discussed and, ideally, included as baselines.

- The UCB/TS exploration terms depend on problem-dependent quantities (like the RKHS norm bound and noise level). These are not observable in practice, so the theoretical guarantees rely on tuning parameters you don’t actually know.

- While the bound is written in terms of an “effective dimension” $\tilde{d}$, the asymptotic rate is the same $\sqrt{T}$ scaling as standard kernelized bandits. The paper does not prove matching lower bounds or show regimes where $\tilde{d}$ is provably small, so the improvement can be marginal.

**Questions:**

See weakness.

---

> ### Author Response · Authors · 2025-11-26
>
> We appreciate your valuable suggestions. Please see our responses to the four points you raised.
>
> > ... asymptotic rate is the same $\sqrt{T}$ scaling as standard kernelized bandits ...
>
> We thank the reviewer for this legitimate and insightful concern. It motivated us to significantly deepen our theoretical analysis, leading to the addition of **Section 4.4 (Spectral Analysis of the Multi-User Kernel)** in the revision. This new analysis explicitly characterizes the effective dimension $\tilde{d}$ based on the graph spectrum and reveals a much richer story than originally presented.
>
> **1. Scaling with $n$ and the "Standard" Rate:**
> While a $\sqrt{T}$ rate is standard for single-agent kernel bandits, the key distinction in the multi-user setting is the scaling with $n$.
> *   **Independent Baseline:** Without graph information, $n$ independent learners achieve a collective regret of $\sum_{u=1}^n \sqrt{T/n} = \sqrt{nT}$. In the data-scarce regime where $T \asymp n$, this implies linear regret $\mathcal{O}(T)$, i.e., no learning occurs.
> *   **Our Contribution:** We proved that under strong homophily (e.g., a clique or $k$ clusters), our effective dimension $\tilde{d}$ saturates and becomes independent of $n$. Consequently, our algorithm achieves $\mathcal{O}(\sqrt{T})$ regret even when $T \asymp n$. This is not a marginal improvement; it is the difference between learning nothing and learning efficiently.
>
> **2. Provably Small Effective Dimension (New Section 4.4)**
> In the revision, we provide a rigorous bound on $\gamma_T$ using a "regular design" argument (Theorem 1 implies a tensor product structure). We analyze two distinct regimes:
> *   **Worst Case (Independent Users):** We show $\tilde{d} \propto n$.
> *   **Best Case (Strong Homophily):** We prove that for a complete graph, the "tail" eigenvalues of the multi-user kernel collapse. Surprisingly, we show that in the regime $T \le Cn$, the information gain is $\mathcal{O}(1)$, implying that $\tilde{d}$ actually **decreases** as $\mathcal{O}(1/\log T)$. This behavior has no counterpart in the single-user setting and proves that the "price of $n$" vanishes under strong homophily.
>
> **3. Tightness**
> We also added a discussion clarifying that our $\tilde{\mathcal{O}}(\tilde{d}\sqrt{T})$ bound is tight for infinite action spaces (matching linear bandit lower bounds). We further note that for finite action spaces, existing algorithms (e.g., Valko et al., 2013) applied to our new kernel would achieve $\tilde{\mathcal{O}}(\sqrt{\tilde{d}T})$, effectively matching the optimal $\sqrt{nT}$ rate in the independent case while preserving the $\mathcal{O}(\sqrt{T})$ gain in the graph-structured case.
>
> > ... The paper omits key related work. ...
>
> We appreciate the feedback. Unfortunately, the specific citations [1] and [2] were missing from the review text; we would be happy to incorporate them if identified. We note that our original submission did discuss the closest collaborative bandit works in the Appendix. In the revision, we have significantly expanded this discussion and moved it to **Appendix A**, adding detailed comparisons with collaborative and multi-agent bandits. This section now carefully distinguishes our specific setting (user-level homophily) from other graph-structured problems to clearly position our contribution against the closest existing literature.
>
> > ...  exploration terms depend on problem-dependent quantities
>
> We acknowledge that the exploration terms depend on unknown problem-dependent quantities (e.g., RKHS norm, noise variance). However, this dependence is standard across the bandit literature, appearing in the regret bounds of almost all UCB and TS algorithms (e.g., LinUCB [Abbasi-Yadkori et al., 2011], GP-UCB [Srinivas et al., 2009], IGP-UCB [Chowdhury & Gopalan, 2017]).
>
> The theoretical bounds serve to establish the algorithm's optimal regret scaling. In practice, these theoretical confidence widths are known to be overly conservative. Therefore, following standard empirical evaluation practice, we treated the exploration scalar as a hyperparameter and tuned it via grid search. Importantly, this tuning was performed for all algorithms (our proposals and the baselines), ensuring that the reported performance gaps are fair.
>
>
> > ...  baseline comparison is incomplete ...
>
> We appreciate the suggestion but respectfully note that current neural bandit methods are designed for single-agent settings and lack a principled mechanism for Laplacian graph regularization. Our primary goal is to isolate and measure the performance lift provided by multi-user information sharing. To do this rigorously, we must compare against the closest independent variant (GP-UCB). Introducing a neural baseline would confound the benefits of the neural representation with the benefits of graph-based sharing, making it impossible to attribute the gains correctly. Therefore, comparing against graph-agnostic kernel methods is the most precise way to validate our contribution.

---

### Official Review · Reviewer_ZSey · 2025-10-31

**Soundness:** 3
**Presentation:** 3
**Contribution:** 3
**Rating:** 6
**Confidence:** 3

**Summary:**

**Summary**

This paper addresses the multi-user contextual bandit problem where users are related by a graph and reward functions are both non-linear and exhibit graph homophily (i.e., connected users have similar rewards). The authors introduce a joint penalty that combines a graph smoothness term (based on RKHS distances between users) with an individual roughness penalty for each user's function. The central contribution is a proof that this intuitive, additive penalty is mathematically equivalent to the squared norm within a single, unified multi-user RKHS. The authors explicitly derive the reproducing kernel for this new space, which fuses the graph Laplacian and the base arm kernel. This unification allows them to reframe the problem as learning a single "lifted" function, enabling the development of GP-based algorithms. They provide regret bounds that depend on the effective dimension of this new kernel, rather than on the number of users or ambient dimension, and show empirically that their methods outperform strong baselines in non-linear settings.




**Advantages**

* The theoretical unification of the graph smoothness penalty and the individual RKHS penalties into a single, valid multi-user RKHS is the strength of this paper, providing a foundation for the proposed algorithms.

* By deriving the explicit multi-user kernel, the paper bridges the gap between Laplacian regularization and kernelized bandits, allowing the direct application of powerful GP bandit machinery (UCB and TS) to the graph-based problem.

* The resulting regret bounds replace a direct dependency on the number of users ($n$) with a dependency on the "effective dimension" of the new kernel, which captures the spectral properties of both the graph and the arm kernel.




**Shortcoming and Questions**

* The practical implementation of the proposed algorithms requires inverting a $t \times t$ matrix at each step $t$, leading to a computational complexity that scales poorly with the number of rounds, a common issue for GP-based methods.Given that the paper suggests recursive updates as a solution, could the authors provide an experiment comparing the cumulative regret of the exact GP-UCB update versus a more scalable approximation (like the recursive formulas or a sparse GP approach) as the time horizon $T$ becomes very large?


* The derived kernel requires the inversion of the $n \times n$ regularized Laplacian $L_{\rho}$, which could be computationally prohibitive if the number of users ($n$) is extremely large (e.g., millions of users). Would it be possible to conduct an experiment analyzing the scalability of the method with respect to the number of users $n$, and perhaps explore whether an approximation of $L_{\rho}^{-1}$ (e.g., using graph sparsification or spectral methods) could maintain competitive regret?


* The theoretical regret bounds depend on the effective dimension $\tilde{d}$, which is data-dependent and defined based on the entire sequence of actions up to time $T$, making it an a-posteriori quantity that is not known in advance. Could the authors provide an empirical analysis plotting the growth of the effective dimension $\tilde{d}$ relative to $T$ in the synthetic experiments, to give a clearer intuition for how this value behaves in practice?

**Strengths:**

Please see above.

**Weaknesses:**

Please see above.

**Questions:**

Please see above.

---

> ### Author Response · Authors · 2025-11-26
>
> We thank the reviewer for these practical considerations regarding scalability.
>
> **1. Recursive Updates are Exact (Not an Approximation).**
> We wish to clarify that the recursive update formulas derived via the blockwise inversion lemma are _algebraically exact_. They are not approximations. Therefore, the regret of the recursive implementation is mathematically identical to the direct inversion approach (up to floating-point precision). A separate experiment comparing them would yield identical learning curves. In our implementation (detailed in **Appendix F.4**), we use a hybrid strategy: we use stable Cholesky decomposition for small $t$ and switch to exact rank-1 recursive updates for large $t$ to maintain $O(n^2)$ complexity per step without sacrificing accuracy.
>
> **2. Scalability with $n$ (Laplacian Inversion).**
> We acknowledge that naive inversion of the $n \times n$ Laplacian is $O(n^3)$. However, this is not a bottleneck in practice for two reasons:
> *   **Sparse Graphs:** For large sparse networks (e.g., social graphs where the number of edges $m \sim n$), **Fast Laplacian Solvers** [Spielman & Teng, 2004; Kyng & Sachdeva, 2016] can solve these linear systems in $\tilde{O}(m)$ (nearly linear time), allowing our method to scale to millions of users without explicit inversion.
> *   **Dense/General Graphs:** As our new spectral analysis (Section 4.4) shows, under homophily, the multi-user kernel is effectively low-rank. This justifies the use of **Spectral Truncation** (approximating $\mathbf{K}_G$ using the top-$r$ eigenvectors), which we already discuss as an implementation detail in **Appendix F.4**. This reduces complexity from $O(n^3)$ to the cost of a partial eigendecomposition ($O(n \cdot r^2)$).
>
> **3. Empirical Validation of Effective Dimension ($\tilde{d}$).**
> We have included the requested empirical analysis in **Figure 1** of the revised submission.
>
> 1.  **Scaling with $T$:** The plot confirms that the effective dimension grows very slowly with $T$. We note that this behavior is theoretically expected for standard kernels (e.g., Squared Exponential) and is now explicitly derived in the newly added **Section 4.4**.
> 2.  **Scaling with $n$ (New Insight):** The plot also reveals the behavior with respect to the number of users $n$. While the metric scales linearly with $n$ for independent users (worst case), it becomes **invariant to $n$** under strong homophily (best case). This provides strong empirical evidence for our new theoretical result in Section 4.4: that the "price of $n$" vanishes in dense graphs, making the effective dimension a benign and stable quantity in practice.

---

### Official Review · Reviewer_xoe9 · 2025-11-01

**Soundness:** 2
**Presentation:** 3
**Contribution:** 3
**Rating:** 6
**Confidence:** 4

**Summary:**

This paper studies multi-user contextual bandits with a known user graph. The authors show how individual RKHS-based reward modeling and graph homophily (via the graph Laplacian) come together by deriving the reproducing kernel of a unified multi-user RKHS. Using this kernel, they design GP-UCB and Thompson Sampling algorithms and analyze their regret.

**Strengths:**

1. Theorem 2.1 provides a clean theoretical connection between Laplacian regularization across users and RKHS-based regularization on arm features. This connection is elegant and makes the overall framework principled and well-founded.

2. The proposed GP-based UCB algorithm are grounded on solid theoretical foundations. The corresponding regret bounds based on the effective dimension are sharp and well-justified, making the theoretical contribution of the paper clear and convincing.

**Weaknesses:**

1. The formulation mainly builds on previous work on linear Laplacian bandits by extending the idea from linear models to general RKHS functions. While the conceptual novelty is moderate, I think this extension is meaningful and technically non-trivial, as it requires re-deriving the RKHS characterization and associated regret analysis.


2. I am a bit confuse about the result for TS, as described below in detail.

**Questions:**

I am a bit surprised by the reported TS result. With a naive TS design, the paper claims a frequentist regret of order $d\sqrt{T}$. However, up to my knowledge, even in the simpler linear bandit setting, the naive version of TS typically achieves only $d^{3/2}\sqrt{T}$ frequentist regret, and stronger results usually require additional design modifications such as the feel-good TS algorithm [1]. Could the authors clarify what differs in their analysis that allows this improved bound? Or are there extra assumptions that effectively bypass the limitations known in previous TS literature?



[1] Zhang, Tong. "Feel-good thompson sampling for contextual bandits and reinforcement learning." SIAM Journal on Mathematics of Data Science 4.2 (2022): 834-857.

---

> ### Author Response · Authors · 2025-11-26
>
> > ... The formulation mainly builds on previous work on linear Laplacian bandits  ...
>
> We thank the reviewer for recognizing the technical value of our extension to the RKHS setting.
>
> We would like to highlight that we have provided a thorugh analysis of the effective dimension in our revision, revealing that this extension offers more than just a broader function class: it fundamentally alters the regret scaling properties in ways not easily visible in the linear case. In the newly added **Section 4.4**, we derive a spectral bound showing that under strong homophily, the effective dimension $\tilde{d}$ of our multi-user kernel is **independent of the number of users $n$**, and in fact **decreasing slowing** in $T$, in the regime $T \asymp n$. As a result the regret is **independent of $n$** in this regime as well.
>
> This implies that the "price of $n$" vanishes entirely for dense graphs, a surprising result that has no counterpart in single-user settings and, to our knowledge, has not been explicitly characterized in prior multi-user bandit work. We refer to our response to **Reviewer uAXP** for a detailed summary of these new spectral findings.
>
> > ... TS rate
>
> We thank the reviewer for this sharp observation.
>
> The reviewer is correct that for infinite action sets, the standard frequentist regret for Thompson Sampling scales as $\tilde{O}(d^{3/2} \sqrt{T})$. However, Agrawal & Goyal (2013) [1] explicitly establish that restricting the problem to **finite action sets** (with $N$ arms) allows one to "shave off" a factor of $\sqrt{d}$ and replace it with $\sqrt{\log N}$.
>
> Our improved rate relies on two factors:
> 1.  **Finite Actions:** We operate in the finite-action regime, which allows replacing the extra $\sqrt{d}$ factor with logarithmic terms.
> 2.  **Variance Inflation:** Our theoretical analysis follows the standard "frequentist analysis of Bayesian updates" framework for kernel bandits (e.g., [Chowdhury & Gopalan, 2017; Zhang et al., 2020]). Specifically, our algorithm (Alg. 2) and analysis incorporate an inflation parameter $\nu_t$ for the posterior variance. By setting $\nu_t$ to scale with the confidence width $\beta_t$, the TS algorithm effectively mimics the exploration envelope of UCB, allowing us to recover the $\tilde{d} \sqrt{T}$ rate.
>
> We also note that originally both our UCB and TS bounds were stated under an umbrella finite action set assumption presented early in the initial submission. Since our UCB bound holds generally for infinite action sets, we have now separated the assumptions in the revision: we explicitly state in the theorems that our UCB bound holds under infinite action sets, while the TS bound relies on the finite action set assumption.
>
> [1]  Shipra Agrawal, Navin Goyal, Thompson Sampling for Contextual Bandits with Linear Payoffs

---

### Official Review · Reviewer_jACQ · 2025-11-03

**Soundness:** 3
**Presentation:** 3
**Contribution:** 3
**Rating:** 6
**Confidence:** 4

**Summary:**

This paper addresses the problem of multi-user contextual bandits and proposed a framework based on a novel penalty that explicitly captures the relations between the user reward functions. The authors went on to provide a justification of this penalty by linking it to a multi-user RKHS, which allows for the development of two Gaussian process based algorithms. Theoretical result on  the regret bound has been derived for both algorithms, and experimental results validate their  effectiveness and superiority over existing baselines.

**Strengths:**

- The paper tackles an under-explored but interesting problem.
- The formulation of the penalty and its link to a RKHS is interesting and provides guidance on how relational information can be taken into account in learning.
- Theoretical results on regret bounds are a plus.
- The paper is generally well presented with clear motivation and technical descriptions.

**Weaknesses:**

**Connection to existing literature.** I believe it would be helpful for the authors to make a greater effort connecting their approach to similar attempts in the literature:
- First, the link between kernels and regularisations, as well as its extension to the graph case via the graph Laplacian, has been well-documented in the literature [1], and in my view this should be properly discussed in the derivation of the proposed approach.
- Second, the way the graph structure is incorporated into learning bears similarly with recent work in the space of multi-output and graph-based GPs, where this is done either implicitly [2] or explicitly [3,4]. I feel additional discussion and comparison against those would make the contribution of the paper conceptually clearer.

**Technical framework.** The paper only focuses on a smoothness assumption, which is a reasonable starting point, however I was left wondering whether this can be generalised to the case where the user reward relations are more complex? I feel this should be possible to address by following for example the spectral learning approach in [4].

**Experiments.** Experimental results, while convincing, are relatively brief:
- Can the authors demonstrate the impact of different graph topology on learning performance, for example by using graphs generated from the stochastic block model? I think this can connect nicely to the discussion on effective dimension.
- Can the authors provide ablation studies that demonstrate the utility of both the graph-based kernel and the classical kernel in the Kronecker product?
- All experiments are synthetic at the moment, can the authors think about potential real-world experiments? This can also help a reader appreciate the practical utility of the proposed approach.

[1] Smola and Kondor, “Kernels and Regularization on Graphs,” COLT, 2023.

[2] Alvarez et al., “Kernels for vector-valued functions: A review,” Foundations and Trends in Machine Learning, 2012.

[3] Venkitaraman et al., “Gaussian processes over graphs,” IEEE ICASSP, 2020.

[4] Zhi et al., “Gaussian Processes on Graphs Via Spectral Kernel Learning,” IEEE TSIPN, 2023.

**Questions:**

See weaknesses above for the specific points I would like the authors to address or discuss.

---

> ### Author Response · Authors · 2025-11-26
>
> We appreciate valuable suggestions from the reviewer.
>
> > ... Connection to existing literature
>
> We thank the reviewer for pointing out these fundamental references, which we have now incorporated to better contextualize our work. Please see the expanded **Appnedix A** on related work (this used to be Appnedix D in the intial submission). We also added clear pointers to this appendix in the main text.
>
> **1. Connection to Smola & Kondor [1]:**
> We agree that [1] is the foundational work establishing the link between the discrete Laplacian norm and the inverse Laplacian kernel. We have updated the paper to explicitly credit [1] for the scalar case ($K_x \equiv 1$). Our Theorem 1 can be viewed as the vector-valued extension of their result, generalizing the smoothing from scalar graph signals to functions over user-context pairs via the tensor product structure.
>
> **2. Connection to Multi-output and Graph GPs [2,3,4]:**
> We have added a discussion comparing our approach to these works.
> *   **Alvarez et al. [2]:** We acknowledge that this work anticipates the tensor product structure $K_{\text{task}} \otimes K_{\text{input}}$ for vector-valued functions and notes the inverse relationship between the regularization matrix and the kernel. Our work rigorously applies this structure to the specific Gang-of-Bandits objective.
> *   **Venkitaraman [3] & Zhi [4]:** While these works also utilize tensor product kernels involving the Laplacian, they often focus on learning general spectral filters (e.g., polynomial filters in [4]). In contrast, we fix the graph kernel to the Green's function $(\mathbf{L}+\rho \mathbf{I})^{-1}$. This is a deliberate choice: it maintains a strict equivalence to the specific graph-homophily regularization term (the Dirichlet energy) used in the bandit literature, allowing us to derive regret bounds based on the effective dimension of this specific norm-induced RKHS.
>
> > ... The paper only focuses on a smoothness assumption,
>
> This is an interesting point. The smoothness assumption we adopt is best viewed as a homophily assumption: users that are close in the graph are encouraged to have similar reward functions, where the notion of similarity is itself quite general and is determined by the choice of kernel and the induced RKHS norm. In that sense, our framework can already encode a wide range of user–reward relations, subject to the constraint that they be "normable" in a Hilbert space.
>
> The learnable spectral approach in [4] goes in a different direction by parameterizing the graph component as a polynomial in $\boldsymbol L$ rather than through $(\boldsymbol L+\rho \boldsymbol I)^{-1}$. While this may be attractive in practice, it is harder to interpret from first principles: working directly with polynomials of $\boldsymbol L$ makes the induced structure on the reward functions opaque, and, since Laplacian-based constructions are themselves closely tied to smoothness, it is not clear that this fundamentally changes the type of relations being imposed. Moreover, optimizing over spectral filters leads to a nonconvex problem, which complicates regret analysis. Designing alternative, theoretically well-justified models of user–reward relations beyond such norm-based smoothness remains an open and interesting direction.
>
> > ...  impact of different graph topology ...
> > ...  the utility of both the graph-based kernel and the classical kernel in the Kronecker product?
>
> In response to these suggestions, we have added a new Section~4.4 that explores the effective dimension both theoretically and empirically. In particular, we show that under strong homophily the effective dimension is significantly reduced: when $T \asymp n$, it actually decreases on the order of $1/\log T$ and is independent of $n$. This directly illustrates the benefit of the Kronecker product structure: the graph kernel and the context kernel act together to reduce the effective dimension beyond what is possible in the single-user setting.
>
> We also discuss the case of $k$ strongly separated clusters, which is essentially similar to a $k$-class SBM. In this regime, the effective dimension is multiplied by $k$ relative to the strong-homophily (clique) case, rather than by $n$, making explicit how the graph topology controls learning complexity. Complementing this theory, Figure 6 in Appendix F now reports empirical results for three random graph topologies, including an SBM.
>
> > ... real-world experiments?
>
> Regarding real-world experiments, suitable multi-user bandit datasets with an explicit user graph are, to the best of our knowledge, not readily available. Constructing such a dataset is beyond the scope of this work, so we follow the standard practice in this line of research and focus on controlled synthetic experiments.

---

> ### Comment · Reviewer_jACQ · 2025-11-27
>
> I thank the authors for their responses to my previous comments, which helped improve the paper. There are a few points that I would still like to mention:
> - I believe the work of Smola and Kondor (2003) is too relevant to be only discussed in Appendix. One of the key contribution of the paper is to establish a precise link between the Laplacian based regularisation and the corresponding kernel function, and it laid a foundation to what the paper is proposing. Therefore I believe this should be discussed properly in the 3rd paragraph of Introduction.
> - The authors stated that "working directly with polynomials...type of relations being imposed," which I respectfully disagree. First, due to the precise link between the Laplacian based regularisation and the corresponding kernel function, there is a clear interpretation of any valid kernel function from the regularisation viewpoint, and a nonsmooth assumption would simply correspond to an $r(\lambda)$ function in Smola and Kondor (2003) that is not monotonically increasing. Second, polynomials of Laplacians can capture a wide range of behaviour, so they do not have to be related to a smoothness assumption. I think it would be nice to have a properly discussion about the smoothness assumption the paper is making and possibilities to go beyond it.
> - The new Section 4.4 is welcome. Can I ask why a complete graph would necessarily correspond to strong homophily though? I believe the structure itself is not sufficient to determine this and we need additional assumption on the reward functions.
> - I understand there is a lack of real-world datasets that are suitable for the paper's setting, however this makes it somewhat difficult to evaluate the practical utility of the work. Can the authors at least mention a few practical scenarios where the proposed method might be helpful?

---

> > ### Author Response · Authors · 2025-12-03
> >
> > We appreciate quick and constructive feedback from the reviewer. Below we address each point and summarize the corresponding changes in the revised version.
> >
> > ---
> >
> > **1. Smola & Kondor (2003) in the Introduction**
> >
> > We completely agree and have revised the **third paragraph of the Introduction** to explicitly acknowledge Smola and Kondor (2003) as the foundational scalar case.
> >
> > - In the revised text, we now state that in the scalar setting without contextual features, Laplacian-based regularization is known to induce a kernel matrix equal to the (regularized) Green’s function of the graph, as shown by Smola & Kondor.
> > - We then position **Theorem 1** as a vector-valued / multi-user extension of this result: the same regularization principle is lifted from scalar graph signals to functions over user–context pairs, yielding a joint kernel
> > $K((x,u),(x',u')) = [L_\rho^{-1}]_{u,u'}\,K_x(x,x')$.
> >
> > This makes the dependence on their work explicit in the main narrative and clarifies how our contribution builds directly on their framework.
> >
> > ---
> >
> > **2. Smoothness assumption and going beyond it (spectral viewpoint)**
> >
> > We thank the reviewer for pointing out that our earlier wording was too strong. We have revised both the **discussion of related work** and the **spectral interpretation** to better reflect the generality of Laplacian-based kernels.
> >
> > 1. **Spectral characterization in Appendix A.**
> >    In **Appendix A (Related Work)** we now explicitly adopt the spectral viewpoint of Smola & Kondor and write any Laplacian-based kernel in the form
> > $K_G = \sum_{i=1}^n r(\lambda_i)\, q_i q_i^\top$, where \((\lambda_i, q_i)\) are the eigenpairs of \(L\) and \(r(\cdot)\ge 0\) is a spectral transfer function.
> > Our choice \($K_G = (L+\rho I)^{-1}$\) corresponds to
> > $ r(\lambda) = \frac{1}{\lambda + \rho}$, which is monotone decreasing in \(\lambda\) and thus penalizes high-frequency Laplacian modes more strongly. This is exactly the classical **homophily/smoothness** assumption on the user side.
> >
> > 2. **Clarifying our earlier statement about \($g(L)$\) / polynomials.**
> >    We have rewritten the paragraph discussing Venkitaraman et al. (2020) and Zhi et al. (2023) to avoid the implication that kernels based on \($g(L)$\) or polynomials of \($L$\) are “opaque.” The revised text now:
> >    - acknowledges that the underlying regularizer can indeed be described spectrally in terms of a transfer function \($r(\lambda)$\) in the sense of Smola & Kondor;
> >    - clarifies that our methodological choice is to **commit to the specific Green’s-function kernel** \($K_G = (L+\rho I)^{-1}$\), which corresponds to the Dirichlet-energy regularizer and yields a particularly simple and interpretable norm that we can track explicitly in our bandit analysis.
> >
> > 3. **Possibilities beyond smoothness.**
> >    In the same appendix, we now explicitly note that non-monotone or band-pass transfer functions \($r(\lambda)$\) can encode more complex, possibly non-smooth or heterophilous relations between users. We highlight this as a promising direction for future work: our present regret analysis focuses on the homophilous, monotone-smooth regime, but the spectral framework is general and can, in principle, accommodate richer priors.
> >
> > We hope this revised discussion better reflects the flexibility of Laplacian-based kernels while clearly stating the specific homophily assumption under which our current theory is developed.
> >
> > ---
> >
> > **3. “Complete graph” versus “strong homophily” in Section 4.4**
> >
> > We agree that graph structure alone is insufficient; however, the necessary constraint on the reward functions is already provided by **Assumption 3 (Bounded Multi-User RKHS Norm)**. Recall that the squared norm in our unified RKHS (Eq. 1) is defined as:
> >
> > $\|f\|_{\mathcal{H}}^2 = \rho \sum_{i} \|f_i\|_{\mathcal{H}_x}^2 + \frac{1}{2} \sum_{i,j} w_{ij} \|f_i - f_j\|_{\mathcal{H}_x}^2 \leq B_{\rho}^2$.
> >
> > For a "Complete Graph" (where $w_{ij} > 0$ for all pairs), the dominance of the second term (the Dirichlet energy) implies that for the total norm to remain bounded by $B_{\rho}^2$, the pairwise differences $\|f_i - f_j\|_{\mathcal H_x}^2$ must be small for all pairs.
> >
> > Thus, **Assumption 3 is sufficient**: imposing a bounded norm on a dense graph mathematically forces the reward functions to be strongly homophilous. This is the advantage of our unified RKHS framework: the graph structure and the function regularity are coupled in a single constraint.
> >
> >
> > **4. Practical utility and real-world scenarios**
> >
> > We appreciate this suggestion. While we still do not have access to a real-world dataset with a known user graph and a shared arm catalog in a bandit setting, we have now added a paragraph on **practical scenarios** in the experimental supplement.
> >
> > ---
> >
> > We hope these clarifications and textual changes address the reviewer’s concerns and make the scope, assumptions, and potential applications of our work clearer.

---

### Meta-Review · Area_Chair_pmYu · 2026-01-05

**Summary:**

All reviewers like the solid theoretical foundations of this work, for example, the clean theoretical connection between Laplacian regularization across users and RKHS-based regularization on arm features. Major concerns raised by reviewers lie in missing some related work, presentation of some theoretical results, and incomplete experiments, and they asked for further clarifications and suggested some additional experiments. After reading the author rebuttal, the discussion between authors and Reviewer jACQ, and the paper revision, I find that all concerns can be addressed by clarifications and newly-added experiments. Therefore, a unanimous decision towards accept can be achieved, and I recommend accept.

**Reviewer Concerns:**

Major concerns raised by reviewers lie in missing related work, theoretical results, and incomplete experiments. Most concerns can be addressed by the author rebuttal.

**Reviewer Scores:**

Reviewer jACQ kept their score at 6 after reading the author rebuttal, but didn't reply after the authors' second response. Unfortunately, Reviewers xoe9, ZSey, and uAXP didn't reply to the author rebuttal. After carefully reading the author rebuttal, I find all concerns can be addressed, so a unanimous decision towards accept can be achieved.

---

### Decision · Program_Chairs · 2026-01-26

Accept (Poster)